# Meta-analysis shows no consistent evidence for senescence in ejaculate traits across animals

Krish Sanghvi [1,4] ✉, Regina Vega-Trejo[1,4] ✉, Shinichi Nakagawa [2], Samuel J. L. Gascoigne [1], Sheri L. Johnson [3], Roberto Salguero-Gómez[1], Tommaso Pizzari[1] ✉ & Irem Sepil [1] ✉

Male reproductive traits such as ejaculate size and quality, are expected to decline with advancing age due to senescence. It is however unclear whether this expectation is upheld across taxa. We perform a meta-analysis on 379 studies, to quantify the effects of advancing male age on ejaculate traits across 157 species of non-human animals. Contrary to predictions, we find no consistent pattern of age-dependent changes in ejaculate traits. This result partly reflects methodological limitations, such as studies sampling a low proportion of adult lifespan, or the inability of meta-analytical approaches to document non-linear ageing trajectories of ejaculate traits; which could potentially lead to an underestimation of senescence. Yet, we find taxon-specific differences in patterns of ejaculate senescence. For instance, older males produce less motile and slower sperm in ray-finned fishes, but larger ejaculates in insects, compared to younger males. Notably, lab rodents show senescence in most ejaculate traits measured. Our study challenges the notion of universal reproductive senescence, highlighting the need for controlled methodologies and a more nuanced understanding of reproductive senescence, cognisant of taxon-specific biology, experimental design, selection pressures, and life-history.

Senescence is the age-dependent irreversible deterioration of organismal function that leads to an increased risk of intrinsic mortality[1] and a decline in reproductive output[2] with advancing age. While senescence has been reported in some taxa[3], it is unclear whether senescence is a general outcome of ageing[4–8]. Senescence is driven by a variety of proximate mechanisms, from excessive biosynthesis in late-life (hyperfunction theory[9]) and age-dependent deterioration of cellular repair[10], to the accumulation of mutations[11], oxidative damage[12], and telomere attrition[13]. From an evolutionary perspective, senescence is commonly hypothesized to be the result of relaxed selection against deleterious mutations in older organisms, as first proposed by Medawar's 'mutation accumulation theory'[14]. Other evolutionary explanations for senescence include selection for alleles, which increase performance early in life but convey net costs later in life ('antagonistic pleiotropy'[15]), and trade-offs between investment in survival versus reproduction ('disposable soma'[16]). In contrast, some animals show an absence of reproductive senescence[4]. Negligible senescence is predicted in animals with indeterminate growth, like some fish[17,18], where individuals continue to grow post-maturity, thus improving their ability to reproduce throughout their lives due to age-dependent

[1]Department of Biology, University of Oxford, Oxford, UK. [2]Evolution and Ecology Research Centre, School of Biological, Earth and Environmental Sciences, University of New South Wales, Sydney, Australia. [3]Department of Zoology, University of Otago, Dunedin, New Zealand. [4]These authors contributed equally: Krish Sanghvi, Regina Vega-Trejo. ✉e-mail: krishsangvi2007@gmail.com; regina.vegatrejo@bms.ox.ac.uk; tommaso.pizzari@biology.ox.ac.uk; irem.sepil@biology.ox.ac.uk

increases in gonad size[8,17]. The ability of some animals to maintain cellular repair and sustain homeostasis in reproductive tissues throughout life might also lead to negligible senescence[19].

Reproductive senescence (i.e., the age-dependent decline in reproductive success) has been relatively well documented in females[20,21]. Yet, patterns, causes, and consequences of male reproductive senescence are less understood[22]. Understanding male reproductive senescence is crucial for several reasons. Males typically face intense intra-sexual competition. Thus, age-dependent changes in male ejaculate traits can drive variation in male reproductive success[23,24], affecting sperm competition, cryptic female choice[25], and generating potential for sexual conflict[26–28]. Additionally, sperm are potentially more vulnerable to organismal ageing than eggs[10,29] because male germlines have higher rates of cell divisions and mutation accumulation[30,31] but poorer DNA repair machinery[12,32] than female germlines. Such deterioration in the male germline can severely impact offspring phenotypes via paternal age effects, thus having important consequences for organismal health (reviewed in[33]).

Current evidence for senescence in male ejaculate traits is inconclusive. Several studies show that older males have lower ejaculate quantities[34] and poorer sperm quality[35–37] than younger males. However, other studies have reported improvements[38–41], or no significant changes in ejaculate traits with advancing male age[42–45]. The heterogeneity in these reported effects might be caused by various biological and methodological factors that modulate the effects of advancing male age on ejaculate traits[35,46] (Tables 1 and 2). A meta-analytical approach is thus crucial to understand the influence of these 'moderators' (Tables 1 and 2) and to investigate the general effects of advancing male age on ejaculate traits. Yet, no study has done this systematically for non-human animals (see[35] for humans;[18] for a review in fish;[47] for effects of male age on seminal fluid).

Here, we conduct a meta-analysis to address three aims. First, we test whether advancing male age affects ejaculate traits across non-human animals (aim 1). Although reproductive senescence is not a ubiquitous outcome of ageing, it is commonly predicted to occur by classical theories of ageing. We thus predict that senescence in ejaculate traits will be observed commonly across species (see Tables 1 and 2 for predictions as to how different ejaculate traits might be affected differently). Second, we investigate the role of biological and methodological moderators (see Tables 1 and 2 for the possible influence of each) in modulating the effects of male age on ejaculate traits (aim 2). Third, we quantify how advancing male age affects reproductive outcomes, such as male fertilisation success and fecundity. Here, we also test whether the effects of advancing male age on ejaculate traits differ from those on reproductive outcomes (aim 3). We find no consistent evidence for senescence in ejaculate traits overall; however, we find taxonomic class- and trait-specific patterns. We also find that studies sampling higher proportions of species' lifespans show stronger evidence for senescence. Overall, we suggest methodological improvements and provide novel hypotheses for studying senescence. The research gaps highlighted by us will be key in aiding our understanding of male reproductive senescence.

## Results

Using a systematic review, we identified 379 studies with relevant data on how advancing male age affects ejaculate traits (Supplementary Fig. 1). From these studies, we obtained 1814 effect sizes across 157 species of non-human animals. We then created a meta-analytical model, using Zr (Fischer's z-transformed correlation coefficient) as our effect size, to understand the overall effects of advancing male age on ejaculate traits. For all our meta-analytical models, we included effect size, cohort, study, species, and phylogenetic relatedness as random effects. From the included studies, we further collected data on various biological and methodological variables (moderators) to test their independent and additive influence on patterns of ageing in ejaculate traits, using meta-regressions. Importantly, for four over-represented taxonomic classes (Mammalia, Insecta, Aves, Actinopterygii), we further conducted four separate meta-regressions to investigate the extent of senescence in ejaculate traits. Some studies also contained additional data on age-dependent changes in reproductive outcomes (e.g. fertilisation success, reproductive output, offspring traits). For these studies, we compared the effects of advancing male age on ejaculate traits and reproductive outcomes. Furthermore, we conducted several analyses to test for different forms of publication biases. Finally, we also conducted two sensitivity analyses to test whether

## Table 1 | Possible influence of different biological moderators on male reproductive senescence at the level of ejaculate traits

| Biological moderators | Possible influence |
|---|---|
| Taxon-specific effects* | Phylogenetic history and taxa-specific biology (e.g. ecosystems, niches, metabolic rates, mating systems, mode of thermoregulation, degree of parental care) could influence how male age affects ejaculate traits[4,59]. |
| Ejaculate traits* | Evidence for reproductive senescence can depend on the specific trait measured[61]. This can be due to trade-offs between different ejaculate traits[60] or different traits being under varying selection pressures[62]. |
| Degree of sperm competition* | Species with increased levels of sperm competition have evolved increased investment in competitive ejaculate traits such as sperm number and velocity[95], which may reduce the rate of senescence in these traits[42]. However, high levels of sperm competition may also lead males to produce large, high-quality ejaculates early in life but exacerbate senescence in ejaculate traits at older ages[96]. |
| Life-history strategies and mortality risk | Life-history strategies of animals and the pace of life of individuals determine the rate and onset of reproductive senescence[48]. Life-history strategies are affected by mortality risk in populations. For instance, animals may invest more in early-life reproduction when age-dependent mortality risk is high[97] and thus show higher reproductive senescence rates than animals facing lower age-dependent mortality risk[98]. Organisms that evolve in environments with high extrinsic mortality might show faster rates of senescence when old due to deleterious late-life expressed alleles not being selected against[15]. |
| Seminal fluid changes | Levels of antioxidants in seminal fluid[47] and abundance of seminal fluid proteins can change as males age[66], independent of changes in sperm. These age-dependent changes in the seminal fluid can affect sperm phenotype over and above the direct effects of male age on sperm[47]. |
| Ontogeny of secondary sexual traits | The ontogeny of secondary sexual traits can influence the evolution of male reproductive senescence rates[27]. For instance, in species where male traits such as weapons or ornaments improve with age, males are hypothesised to evolve lower rates of reproductive senescence, compared to species where these traits do not improve with age[40,51]. |
| Parental care | Species with parental care might have evolved to allocate more energy/resources to caring for offspring and investing in current reproductive opportunities at the cost of reduced allocation to future reproduction. This could accelerate reproductive senescence in species with parental care[2]. |

Moderators marked with an asterisk were included in our meta-analysis because there were sufficient data across studies.

**Table 2 | Possible influence of different methodological moderators on male reproductive senescence at the level of ejaculate traits**

| Methodological moderators | Possible influence |
|---|---|
| Proportion lifespan sampled* | A higher proportion of lifespan sampled will increase the probability of detecting reproductive senescence, as the onset of senescence usually occurs late in life[4,18,35,47]. |
| Ejaculate collection method* | If males have control over ejaculation during ejaculate collection (e.g. natural mating or mating with dummy females), males might have the opportunity to strategically adjust ejaculate phenotypes[99]. This could cause age-independent changes in ejaculate traits, reducing the detectability of senescence. Additionally, when males have control over ejaculation, studies might obtain a smaller proportion of the sperm reserves available to a male, which may not be representative of a male's whole-ejaculate phenotype, compared to studies that use invasive methods to obtain ejaculates (e.g. dissection). |
| Population type* | Reproductive senescence rates can differ between males in captive versus wild populations[68,100]. Additionally, some domesticated animals are often culled prior to reaching ages where senescence can be detected[101]. Other domesticated animals have undergone generations of artificial selection for unusual life histories (e.g. extremely short generation time in broiler chicken[102]. These factors could lead to patterns of senescence differing between domesticated and wild animals. |
| Cross-sectional versus longitudinal sampling* | A cross-sectional sampling of males makes reproductive senescence harder to detect, especially if low-quality males selectively disappear[55,56]. Cross-sectional studies might thus underestimate male reproductive senescence, compared to the longitudinal sampling of the same males at different ages[103]. |
| Manipulations* | Manipulated environments that are outside of what healthy organisms typically experience, such as environments with stressful conditions, can exacerbate reproductive senescence[104]. Thus, males exposed to manipulations such as thermal stress, poor diet, or toxins could be more likely to show reproductive senescence than males not subjected to these stressors. Other manipulations, such as experimental inbreeding[105] or selection for deleterious mutations[106], may exacerbate reproductive senescence. |
| Mating history | High mating rates can exacerbate male reproductive senescence[23]. In studies where male mating history is not controlled for, old males often have more matings than young males. These studies might thus show stronger evidence for senescence in ejaculate traits. On the other hand, low mating rates (e.g. virgins) might cause old males to accumulate sperm for longer durations, thus producing larger ejaculates than young males[66]. |
| Post-meiotic sperm storage | Temporal changes in sperm traits can also occur due to post-meiotic storage of mature sperm in males before ejaculation and in females following mating[54]. The duration of sexual rest in males can influence the amount of post-meiotic damage to sperm, such that for a given age, males with shorter sexual rest (e.g. high mating rate) will incur lower post-meiotic sperm damage[54]. Further, deleterious effects of post-meiotic sperm storage may be exacerbated in old males if old males are less able to repair post-meiotic cellular damage in sperm[54]. |

Moderators marked with an asterisk were included in our meta-analysis because there were sufficient data across studies.

evidence for senescence was sensitive to the proportion of lifespan of the associated species a study sampled and the aims of the study.

## Aim 1: Effects of advancing male age on ejaculate traits
We found no general effect of advancing male age on ejaculate traits (mean [95% confidence interval (CI)]: −0.006 [−0.486 to 0.474], $z = −0.025$, $P = 0.978$, Fig. 1A). Heterogeneity in our dataset was high ($I^2 = 95\%$), with 40% attributed to true differences between studies, 19% to differences between effect sizes, 0% to between-species differences, and 0.6% to differences between cohorts. Notably, phylogenetic relatedness (Supplementary Fig. 2) explained 35.4% of heterogeneity, suggesting a phylogenetic signal on male reproductive senescence.

## Aim 2: Role of biological and methodological moderators
We did not find a significant general effect of advancing male age on ejaculate traits in our full model (which included all moderators with data for >75% of effect sizes; mean [95% CI]: −0.197 [−1.496 to 1.103]). However, the included moderators explained a significant proportion of the total heterogeneity in our data ($R^2 = 12.17\%$, $Q_M = 99.606$, $Q_E = 15299.075$, $P < 0.001$, DF = 36).

We did not find evidence for age-dependent changes in ejaculates in any taxonomic class (Fig. 1B for four major classes, Supplementary Fig. 3 for all classes), except in Malacostraca (which showed improvement with advancing male age), when effects were averaged across all ejaculate traits. However, taxonomic class explained a significant proportion of heterogeneity ($R^2 = 8.26\%$, $Q_M = 26.082$, $P = 0.025$, DF = 14). Similarly, when averaged across all taxa, we did not find evidence for advancing male age to affect any individual ejaculate trait significantly. Yet, the ejaculate trait explained a small but significant proportion of heterogeneity ($R^2 = 1.72\%$; $Q_M = 51.287$; $P < 0.001$, DF = 13, Fig. 2A).

We detected taxonomic class-specific effects of advancing male age on individual ejaculate traits. For insects (Insecta, $k = 258$), ejaculate size, quantity of sperm (corrected for body or testis size), number of sperm, and sperm viability, improved with advancing male age (Fig. 2B). For ray-finned fish (Actinopterygii, $k = 174$), sperm motility and velocity decreased, whereas ejaculate size increased, with advancing male age (Supplementary Fig. 4A). However, we found no significant effect of advancing male age on individual ejaculate traits in birds (Aves, $k = 318$; Supplementary Fig. 4B) or mammals (Mammalia, $k = 990$; Supplementary Fig. 4C).

We also observed species-specific effects of advancing male age on individual ejaculate traits. For lab rodents, *Rattus norvegicus* and *Mus musculus* ($k = 373$, combined), most traits (i.e. sperm viability, number, motility, per cent of sperm with morphological defects, sperm concentration, sperm mitochondrial function, sperm DNA and oxidative damage) showed senescence (Fig. 2C; Supplementary Fig. 5A). For bulls (*Bos taurus*, $k = 173$), ejaculate size increased with advancing male age (Supplementary Fig. 5B). For *Gallus spp.* (domestic chicken and red junglefowl combined, $k = 183$), number of sperm and ejaculate size showed senescence (Supplementary Fig. 5C; see Fig. 3 for a summary of all taxa- and species-specific effects). The male gonadosomatic index of a species (GSI: i.e. the ratio of testes to body mass, used as a proxy for the degree of sperm competition) did not modulate how advancing male age affected ejaculate traits ($R^2 = 0.26\%$, $Q_M = 0.786$, $P = 0.375$, DF = 1, Supplementary Fig. 6). Finally, using linear mixed-effects models, we detected some evidence for a quadratic effect of advancing male age on the per cent of morphologically normal sperm, viable sperm, and motile sperm (Supplementary Fig. 7).

Studies sampling a higher proportion of the maximum adult lifespan of a species provided stronger evidence for senescence in ejaculate traits ($R^2_{all} = 0.57\%$, $Q_M = 4.838$, $P = 0.028$, DF = 1, Fig. 4A; see Supplementary Fig. 8 for distribution of lifespans sampled across

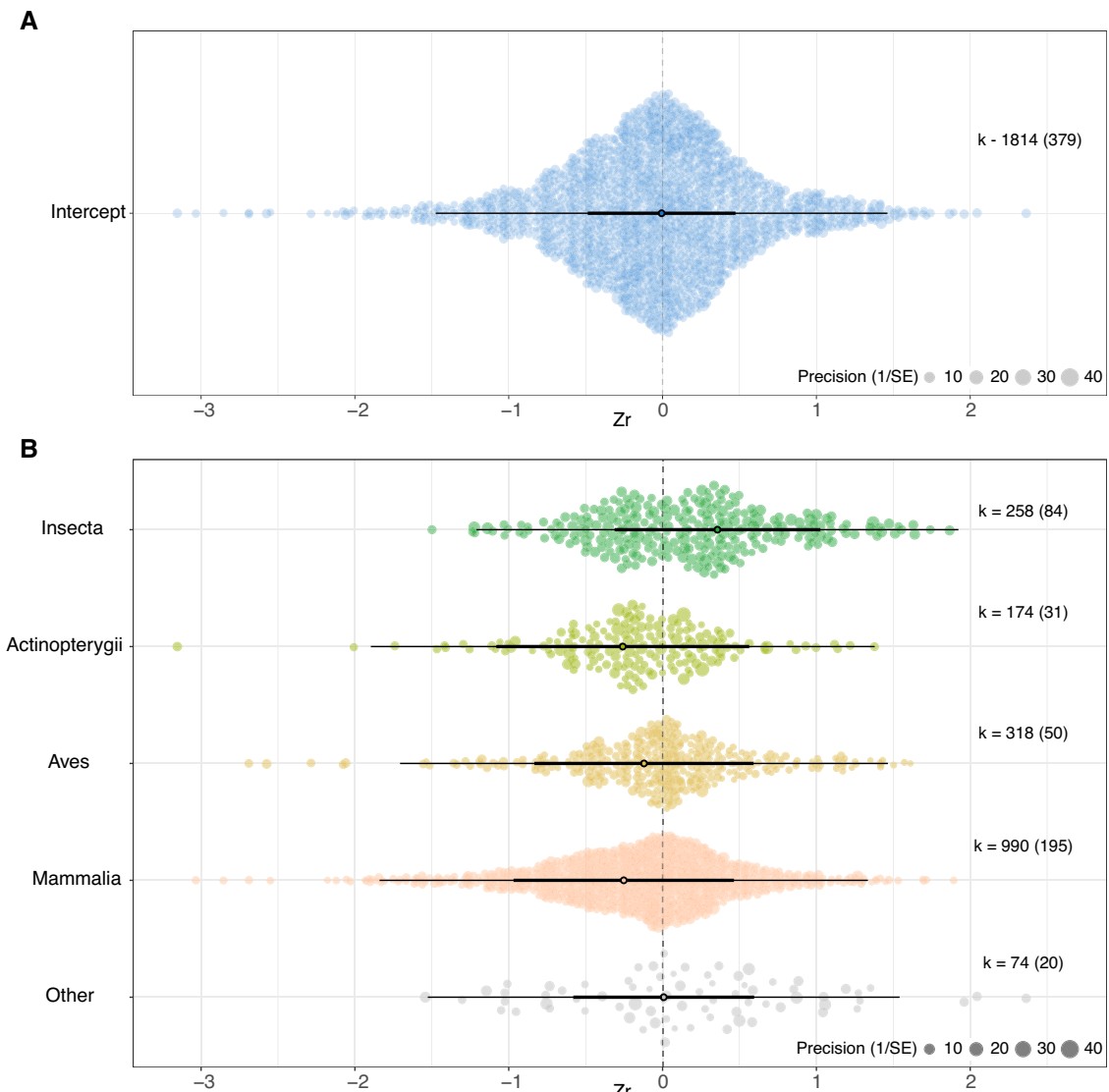

**Fig. 1 | No consistent evidence for senescence in ejaculate traits, irrespective of taxonomic class. A** Meta-analytical model of the overall effect of advancing male age on ejaculate traits. **B** Effect of advancing male age on ejaculate traits for each taxonomic class (note that animal classes with less than 25 effect sizes were grouped together in 'Other'). The size of each data point represents the precision of the effect size (1/SE). The *x*-axis represents values of effect sizes as Fisher's z-transformed correlation coefficient (Zr), while the *y*-axis shows the density distribution of effect sizes. The position of the overall effect is shown by the dark circle, with negative values depicting senescence in ejaculate traits and positive values showing improvement in ejaculate traits with advancing male age. Bold error bars (95% CI) show whether overall effect size is significantly different from zero (i.e. not overlapping zero), while light error bars show the 95% prediction interval (PI) of effect sizes, and black dot shows mean effect size. Sample sizes reported as: *k* = number of effect sizes (in brackets: number of studies). Source data is provided as a source data file.

taxa). This result was supported mainly in captive and lab populations, but not wild and domestic populations ($R^2_{captive}$ = 32.43%, $R^2_{lab}$ = 1.24%, $R^2_{wild}$ = 0.52%, $R^2_{domestic}$ = 0.36%; Fig. 4B–E). The stage of an organism's ontogeny (Supplementary Fig. 9) at which it was sampled significantly influenced the evidence for senescence. Specifically, studies that sampled a higher youngest or oldest age of the associated species (as a proportion of a species' maximum adult lifespan) reported stronger evidence for senescence in ejaculate traits (youngest: *P* = 0.032, $R^2_{all}$ = 0.64%, Supplementary Fig. 10; oldest: *P* = 0.009, $R^2_{all}$ = 0.97%, Supplementary Fig. 11). We did not find evidence for reproductive senescence in ejaculate traits, irrespective of the method used to collect ejaculates from males (e.g. electroejaculation, dissection, natural matings). However, ejaculate collection method explained significant heterogeneity in the data ($R^2$ = 1.36%; $Q_M$ = 7.52, *P* = 0.023, DF = 2, Supplementary Fig. 12). Population type ($R^2$ = 1.12%; $Q_M$ = 2.724, *P* = 0.605, DF = 4, Supplementary Fig. 13) or male sampling method

(i.e. longitudinal or cross-sectional; $R^2$ = 0.08%, $Q_M$ = 0.639, *P* = 0.887, DF = 3, Supplementary Fig. 14), did not modulate the effect of advancing male age on ejaculate traits. We also tested whether males who experienced unnatural manipulations (i.e. conditions outside of their typical range, compared to a well-defined control in the study) showed more senescence than males who did not undergo unnatural manipulations. We detected no senescence or improvement in ejaculate traits irrespective of whether males underwent unnatural manipulations (e.g. heat stress) or not ($R^2$ = 0%, $Q_M$ = 0.021, *P* = 0.989, DF = 2, Supplementary Fig. 15A, B), or found significant differences in effects sizes between manipulated and unmanipulated males (*P* = 0.885).

**Aim 3: Effects of advancing male age on reproductive outcomes**
We found that male reproductive outcomes (i.e. measures of male fertilisation success, reproductive output, or offspring quality) did not improve or decline with advancing male age overall (Supplementary

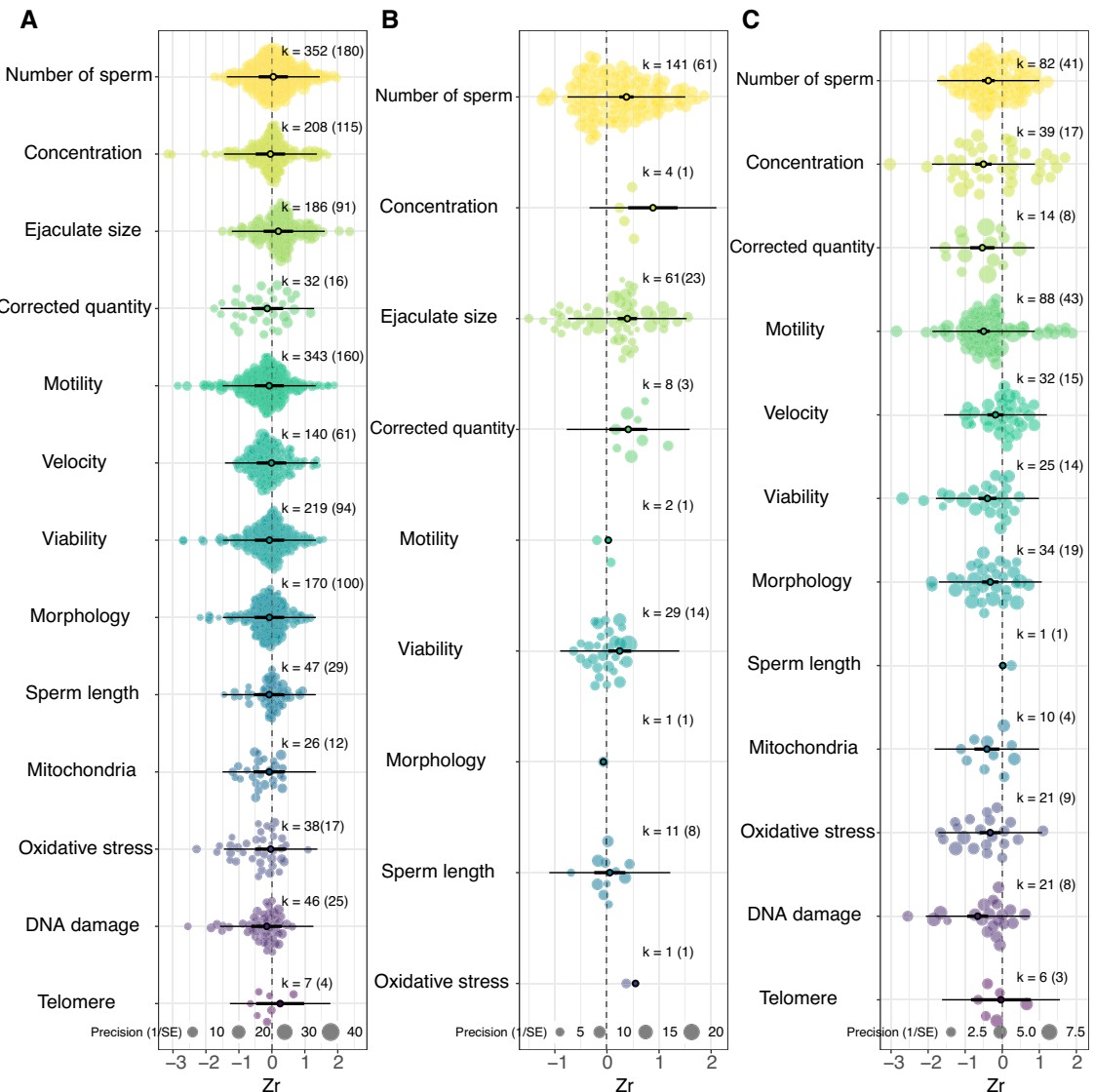

**Fig. 2 | No consistent evidence for senescence in ejaculate traits when all taxa were considered, but some ejaculate traits improve with advancing age (in insects), while other traits decline (in lab rodents). A** Effect of advancing male age on individual ejaculate traits across all 157 species in the dataset. **B** Effect of advancing male age on individual ejaculate traits in the class- Insecta. **C** Effect of advancing male age on individual ejaculate traits for the two most over-represented species combined (lab rodents): *Mus musculus* and *Rattus norvegicus*. The size of each data point represents the precision of the effect size (1/SE). The *x*-axis represents values of effect sizes as Fisher's *z*-transformed correlation coefficient

(Zr), while the *y*-axis shows the density distribution of effect sizes. The position of the overall effect is shown by the dark circle, with negative values depicting senescence in ejaculate traits and positive values showing improvement in ejaculate traits with advancing male age. Sample sizes reported as: k = number of effect sizes (in brackets: number of studies). Bold error bars (95% CI) show whether overall effect size is significantly different from zero (i.e. not overlapping zero), while light error bars show the 95% PI of effect sizes, and black dot shows mean effect sizes. Note that error bars are not provided for traits with a number of effect sizes less than 3. Source data is provided as a source data file.

Fig. 16A). However, reproductive outcomes were less likely to deteriorate with advancing male age, than ejaculate traits ($R^2 = 1.76\%$, $Q_M = 9.783$, P = 0.002, DF = 1; Supplementary Fig. 16B).

## Publication bias

We found no statistical evidence for publication bias, except for a time-lag bias, with more recent studies being more likely to show senescence in ejaculate traits (Supplementary Fig. 17, 18, 19).

## Other sensitivity analyses

We found no significant evidence for senescence in ejaculate traits, even when restricting the analysis to studies that sampled more than 10% of the maximum adult lifespan of the species (mean [95% confidence interval (C.I.)]: −0.020 [−0.549 to 0.509], $z = -0.075$, $P = 0.940$, Supplementary Fig. 20). Results from our taxonomic class-specific

models, which again only included studies that sampled >10% of maximum adult lifespan, were qualitatively similar to results from models that included all studies (Supplementary Fig. 21).

We additionally objectively categorised study aims as explicitly interested in senescence (i.e. studies using "ageing", "ageing", "senescence", "senescent", or "senescing" in their abstracts or titles, $N = 101$ studies) or not ($N = 273$ studies). We did not find significant evidence for overall senescence in ejaculate traits, even when we only analysed studies whose aims were categorised as interested in senescence (mean [95% confidence interval (CI)]: −0.294 [−0.760 to 0.172], $z = -1.238$, $P = 0.216$, Supplementary Fig. 22). Study aims however, explained a significant proportion of heterogeneity in effect sizes ($R^2 = 5.08\%$, $Q_M = 36.287$, $P < 0.001$, DF = 2; Supplementary Fig. 22). Furthermore, studies that were interested in senescence sampled a higher proportion of maximum adult lifespan of the associated species

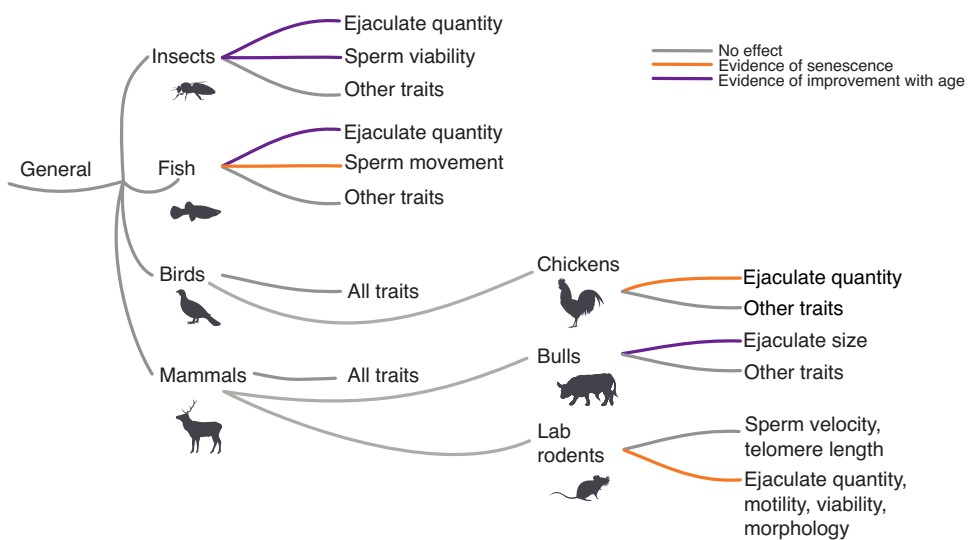

**Fig. 3 | Specific ejaculate traits and taxonomy interacted to affect the evidence for senescence.** Summary of results for how advancing male age affects different ejaculate traits across various taxa in our meta-analysis. "Chickens" refers to domestic chickens and red junglefowl combined. Species icons from PhyloPics, with artist credits and copyright: Kamil S. Jaron (CC0 1.0), Emma Moffett (CC0 1.0), T. Michael Keesey (PDM 1.0), Steven Traver (CC0 1.0), Georgios Lyras (CC0 1.0). Only traits with >3 effect sizes are included in the summary.

(34%) than studies not interested in senescence (20%, Supplementary Fig. 23).

## Discussion

Senescence is central to our understanding of ecology[48], evolution[48], life history[16], and society[49]. Senescence in male ejaculate traits can influence sexual selection[50,51], sexual conflict[26,36], and offspring health[13]. We thus cannot fully understand organismal biology without understanding the evidence for, and consequences of, male reproductive senescence at the level of ejaculates. Our meta-analysis reviews the effects of advancing male age on ejaculate traits across animals in order to test for senescence and highlights key gaps in knowledge that will facilitate a better understanding of ageing.

Contrary to expectations, we detected no consistent evidence for senescence in ejaculate traits across studies (aim 1). Our results contrast those of a meta-analysis in humans[35], which found senescence across most ejaculate traits in men. These differences in results possibly reflect stronger selection pressures in non-human animals to maintain sperm function across all ages compared to men. In our dataset, the phylogenetically closest relative to humans were rodents (exemplified by lab rodents), which, like humans[35], showed evidence for senescence in most ejaculate traits. Current human longevity is much higher than what it was just a few centuries ago[52]. Such recent increases in human longevity could lead to men living much beyond the age at which sperm function can be maintained, leading to greater senescence in the ejaculates of men compared to other animals.

We suggest several potential non-mutually exclusive reasons for the lack of senescence in our meta-analysis. While we discovered that increasing the proportion of lifespan sampled by a study yielded greater evidence for senescence (also shown by[18,35,47]), studies in our meta-analysis tended to sample a low proportion of maximum adult lifespan (median = ~25%, Supplementary Fig. 9), which could have underestimated senescence. Another reason could be that many of the studies included in our analysis were not explicitly testing for senescence. To account for this, we conducted an analysis only on studies that were explicitly interested in senescence. These studies sampled a higher proportion of the lifespan of the associated species yet did not provide evidence for senescence in ejaculate traits overall. However, a reason for this lack of evidence could be that study aims are difficult to quantify, and our classification of aims might have excluded relevant studies. Curvilinear patterns of ageing could also have led us to underestimate senescence. This is because age-dependent changes in ejaculate traits were analysed as a linear function (effect sizes). However, ageing is often curvilinear[4,53]. Our test of quadratic effects showed some evidence in support of this. Thus, if ejaculate traits improve from early to mid-adult life (i.e. maturation) and deteriorate (i.e. senesce) later in life, the positive part of the function would be disproportionately represented against the negative part of the function[54]. Our results overall highlight the need for meta-analysts to develop techniques to calculate and analyse non-linear effect sizes to investigate such patterns.

Selective disappearance of poor-quality males with increasing age could also underestimate senescence[55,56]. Comparing means of age groups in longitudinal studies (like in our meta-analysis) can only account for selective disappearance if all individuals are sampled at all ages, which was rarely done across studies. To account for selective disappearance in cases where not all males survive to be sampled at all ages, we would need to analyse individual-level longitudinal data [rarely reported] for each male in each study rather than comparing the means of different age groups[57]. Age-dependent improvement or negligible senescence in ejaculate traits could also reflect a true biological pattern, with senescence not being an inevitable outcome of ageing for many ejaculate traits and species[4,58]. For instance, taxa with indeterminate growth or slow life histories might show negligible senescence[8,17].

We suggest corollary methodological improvements for more rigorous testing of male reproductive senescence. Specifically, studies could sample higher proportions of a male's maximum lifespan and report survival curves of the studied populations; test for curvilinear effects of age by measuring at least three age cohorts in early-, mid-, and late-adult life; separate confounding effects of male mating history and age by comparing virgin versus frequently mated old and young males; test for selective disappearance by sampling males longitudinally and report individual-level data for each male; be explicit about which theories of ageing are being tested and their corollary predictions; and sample equal number of males in all age classes. Overall, we conclude that senescence is likely occurring in the taxon-specific ejaculate traits where we found supporting evidence. However, we could have underestimated the extent of senescence where

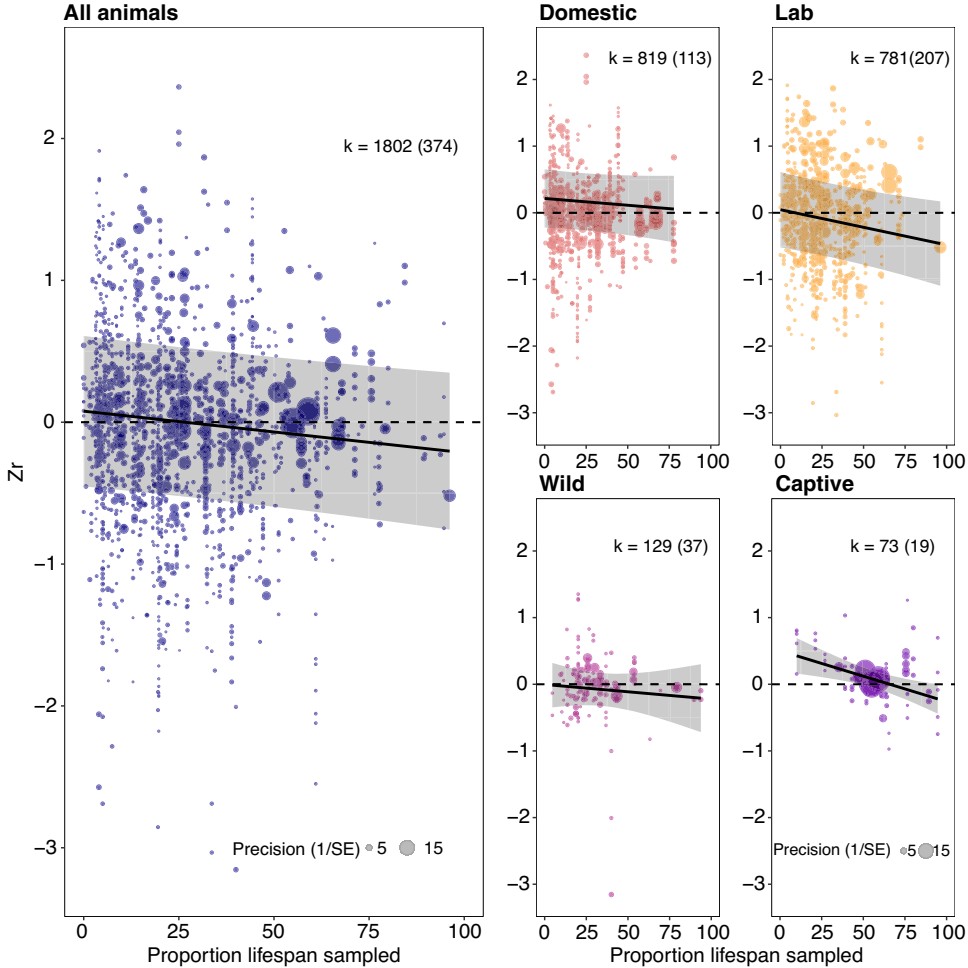

**Fig. 4 | Increasing the proportion of the maximum adult lifespan sampled increased the likelihood of finding senescence.** Effect of proportion of maximum adult lifespan sampled (*x*-axis) on the effect size i.e. Fisher's z transformed r (*y*-axis) across the entire dataset (**A**), and broken down for domestic (**B**), laboratory (**C**), wild (**D**), and captive animals (**E**). The size of each data point represents the precision of the effect size (1/SE). The dark line with shaded bars represents the overall effect of lifespan sampled on effect sizes and its 95% CI, respectively, and the black line shows the mean regression line. Negative values depict senescence in ejaculate traits with advancing age, while positive values show improvement in ejaculate traits with advancing male age. Sample sizes reported as: *k* = number of effect sizes (in brackets: number of studies). Source data is provided as a source data file.

supporting evidence was lacking due to some aforementioned limitations.

Some biological and methodological moderators were important in explaining the observed heterogeneity in effect sizes (aim 2). However, as the effects of these moderators were tested individually, our results could possibly be explained by other moderators not simultaneously included in the analysis. Thus, our results should only be treated as hypothesis-generating rather than evidence of causation. Taxonomic class and ejaculate trait explained a significant proportion of heterogeneity. This heterogeneity could be attributed to differences in ecologies, niches, behaviours, life-history strategies, metabolisms, and evolutionary histories of animals[4,59]. Heterogeneity explained by ejaculate traits could be due to covariances between different ejaculate traits[60]; some traits being more sensitive to age-dependent deterioration than others[61]; or different traits being under varying selection pressures[62]. Additionally, some ejaculate traits are more likely to influence fertilisation success than others[60]. It is thus possible for traits that are more important determinants of fertilisation success to evolve slower rates of senescence than less important traits[63], which future studies could test.

We discovered some taxonomic class-specific evidence for age-dependent changes in individual ejaculate traits. Insects showed an increase in all sperm and ejaculate quantity traits. This increase could

be associated with their mating status, as most studies (>75%) on insects in our meta-analysis kept males as virgins. Specifically, in species with life-long spermatogenesis and low rates of sperm loss (such as some insects[64,65]), low mating rates can result in old males accumulating more sperm and producing larger ejaculates than young males[66]. Ray-finned fish (Actinopterygii) showed evidence for senescence in sperm velocity and motility but also age-dependent increases in ejaculate size. This result could be due to old males producing larger ejaculates to compensate for senescence in sperm performance. Increases in fish ejaculate size could also reflect the effects of continuous post-maturity growth in many fish species[17,18,67], leading to older males having larger gonads. We did not find consistent evidence for senescence in ejaculate traits in mammals or birds.

We detected several species-specific patterns of senescence. Specifically, most ejaculate traits in lab rodents (*Mus musculus* and *Rattus norvegicus* combined) showed senescence, even when only control/wild-type genetic strains were analysed (e.g. C57 for mice, Brown Norway and Sprague Dawley for rats). This could be due to studies on lab rodents usually having equal sample sizes of males in each age cohort, thus possibly limiting bias towards weighting of the positive part (early- to mid-life) of the curvilinear ageing function. Consistent evidence for senescence in lab rodents could also be associated with senescence being exacerbated in lab-adapted

populations[68]. For a more nuanced understanding of such traits by taxon interactions, we suggest that future studies account for age-dependent changes in body and testes size (as covariates), test for post-meiotic senescence of sperm during storage in males, record whether studied species exhibit continuous spermatogenesis and sperm reabsorption; and measure multiple ejaculate traits simultaneously (i.e. sperm quantity and performance/viability), because sperm quantity versus performance traits might be affected by age in different ways.

Extending the proportion of the maximum adult lifespan sampled increased the evidence for senescence in ejaculate traits for a species. This result suggests that the onset of reproductive senescence usually occurs late in life[4,69], and senescence will more likely be detected if studies sample a larger proportion of lifespan. However, this may be biased by the population sampled, as this association was strong in captive and lab animals but not in wild and domestic animals. We did not find evidence for senescence at any level of other methodological moderators (aim 2). This result could be due to the effects of methodological moderators being taxon-specific or being revealed only under interactions with other methodological or biological moderators. The lack of an effect of study methodologies might also be explained by moderators that we did not include in our analyses (Table 2).

We detected no consistent evidence for overall improvement or senescence in reproductive outcomes of males (i.e. measures of fertilisation success, egg/offspring number/viability/quality; aim 3). Our meta-analysis used data on reproductive outcomes only from studies that also measured ejaculate traits, which possibly represents a biased subset of studies on ageing of reproductive outcomes. However, we found that reproductive outcomes were less likely to exhibit age-dependent deterioration than ejaculate traits. This difference could be due to not all ejaculate traits being key determinants of reproductive success (e.g. fertilisation success[70]), and deterioration in some ejaculate traits having little consequence for a male's reproductive outcome[71]. Lower rates of age-dependent declines in male reproductive outcomes could also be due to female-driven effects (e.g. cryptic female choice, reproductive compensation), which might provide a buffer against low-quality ejaculates of old males. For instance, females might be able to eject poor-quality sperm via cryptic female choice[72], or females mated to older males might compensate by investing more resources into provisioning[73]. Additionally, viability selection in old males could purge low-quality male genotypes, leading to old males having higher means and lower variances for reproductive outcomes than young males[46,74]. These results suggest that age-dependent changes in ejaculate traits may not accurately reflect changes in reproductive outcomes. We emphasize that studies should ideally measure ejaculate traits, male reproductive success, and offspring phenotypes to elucidate the fitness consequences of advancing male age.

## Methods

We followed the PRISMA-EcoEvo guidelines for our meta-analysis[75] and conducted statistical analyses in R[76] v 4.1.2. Supplementary figures 1-24) and Supplementary notes 1-12) are provided in the "Supplementary Information" file. Data, model outputs, metadata, code, PRISMA checklist, and pre-registration have all been deposited at OSF (https://osf.io/dk8sq/).

### Search protocol

We conducted a literature search using search strings on SCOPUS and Web of Science on 21st January and 27th March 2021, respectively (see Supplementary Notes 1 for specific search strings). In addition, we conducted a backward and forward search using seven relevant papers related to the topic of our meta-analysis[11,13,22,24,43,54,77]. We additionally

conducted a search for unpublished research using the Bielefeld Academy Search Engine[78]. Finally, we contacted 56 researchers who study the ecology and evolution of male reproductive senescence to ask for unpublished data. Our search resulted in a total of 9412 unique abstracts from published sources and 271 abstracts from unpublished sources (PRISMA diagram: Supplementary Fig. 1). We screened these abstracts in Rayyan[79] and abstrackr[80] using pre-defined selection criteria (see below). We ensured that the screening process was highly repeatable (Supplementary Notes 2).

### Inclusion criteria

For a study to be included in our analysis, some selection criteria had to be fulfilled during the abstract and full-text screening stages. When screening abstracts, the study had to be a research article (not a review, meta-analysis, or case study) on non-human animals written in English and quantifying ejaculate traits in males of different ages. When screening full-texts, the study needed to contain data on the effects of male age on ejaculate traits, non-overlapping age groups of males, and appropriate data for calculation of effect sizes. We only included studies where at least two age groups of adult males could be compared (see Supplementary Notes 3 for our definition of "adults"). We deemed a total of 379 studies (374 from published and five from unpublished sources) appropriate for data extraction based on our selection criteria and included them in our meta-analysis (PRISMA diagram in Supplementary Fig. 1). These studies represented 157 species.

### Data collection

To quantify the evidence for or against male reproductive senescence (aim 1), we collected data on means, standard deviations (SD) or standard errors (SE), the number of males in each age group, and the number of unique males in the study, wherever reported (see Supplementary Notes 4 for formulae used to calculate SD). If we could not obtain means and SD/SE, we noted the "test statistic" (e.g. $t$ from $t$-tests or $R^2$ values) reported in the study from which effect sizes can be easily obtained. We ensured that the data extraction process was highly repeatable (Supplementary Notes 2).

To understand how biological moderators affect patterns of senescence (aim 2), we recorded information on various biological variables from the 379 studies included in the meta-analysis. We recorded the species and taxonomic class of the study organism, and the ejaculate traits measured in the study (see Supplementary Notes 5 for definitions of each trait). The ejaculate traits were either measures of sperm/ejaculate quantity (e.g. sperm concentration, sperm number, and ejaculate volume), sperm performance (e.g. sperm motility, velocity, viability), or intra-cellular measures of sperm quality (e.g. oxidative stress in sperm, DNA damage to sperm, sperm telomere length). Finally, we recorded the gonadosomatic index (GSI, i.e. the ratio of testis mass to body mass, as a proxy for sperm competition[81,82]) for each species, wherever possible (see Supplementary Notes 6; meta-data on OSF https://osf.io/dk8sq/).

To understand how methodological moderators affect patterns of senescence in ejaculate traits (aim 2), we collected data on various methodological variables from included studies (see Supplementary Notes 7). Initially, we recorded the maximum lifespan (male-specific whenever possible or species-specific when male-specific data were not available) and age at adulthood of the species studied (see Supplementary Notes 6). Data on maximum lifespan and age at adulthood, as well as sources of these data, can be found at OSF (https://osf.io/dk8sq/). We then calculated the proportion of maximum adult lifespan sampled for a species in each study (converted to years). Some of the data on maximum adult lifespans (especially for vertebrates) were obtained from large databases/datasets (that often only reported species-level lifespans without reporting the sex of the measured individuals). Thus, these data may not always accurately reflect the

maximum male lifespans of the specific populations included in our meta-analysis.

We also recorded the method of sperm extraction (e.g. electro-ejaculation, natural mating); population type (whether males belonged to wild, domestic, captive or laboratory populations (see Supplementary Notes 8 for definitions)); method for measuring male age (i.e. whether male age was known directly or indirectly estimated from a measure of phenotype); whether the ejaculate was stored in cold conditions (<5 °C, irrespective of the duration of storage) before analysis of sperm performance; and whether the study was experimental or not[18]. In some studies, males underwent "unnatural manipulations" (see Supplementary Notes 9 for detailed definitions). Here, we also recorded whether the data were obtained from males that underwent these "unnatural" manipulations (i.e. males that experienced conditions outside of their typical range that were compared to a well-defined control in the study) or from males that were used as controls in the same study.

We investigated whether advancing male age affects male reproductive outcomes (aim 3) and whether the effects of male age on reproductive outcomes (see Supplementary Notes 5 for definitions) differ from those on ejaculate traits. For this, we collected data on how advancing male age affects male fertilisation success, the number of eggs produced by the mated females, the number of offspring produced by the mated females, egg viability and hatchability; offspring viability, offspring developmental rate and offspring body condition, whenever available in a study (53 studies in total).

## Calculating effect sizes

We used Fisher's z-transformed correlation coefficient (Zr) as the effect size in our meta-analysis[83]. Each effect size was calculated from either standardised mean differences (when two age groups were compared), simulations (when multiple age groups were compared), or test statistics (see Supplementary Notes 10 for formulae used). Effect sizes from these three calculation methods were not significantly different from each other (Supplementary Notes 10; Supplementary Fig. 24); thus, all effect sizes, irrespective of their calculation methods, were analysed together in our models. We corrected all calculated effect sizes (Zr) by a multiplier to obtain the final effect sizes to be used in the analyses (see Supplementary Notes 10) so that negative effect sizes indicated senescence, while positive effect sizes indicated improvement in ejaculate traits with advancing male age.

## Data analysis

We first created a meta-analytical model (i.e. null model) to test for the general overall effect of advancing male age on ejaculate traits (aim 1), using the rma.mv function in the metafor package[84]. We included the effect size (Zr) as our response variable in the null model and random effects of: effect size ID (which represents the residual within-study variance), cohort ID, study ID, and species name to control for non-independence of effect sizes[85]. We also added a correlation matrix quantifying the phylogenetic relatedness of species in our dataset to control for non-independence arising due to shared phylogenetic history and test for a phylogenetic signal[86]. The phylogenetic tree (Supplementary Fig. 2) was built using the packages ape[87] and rotl[88], which use data from the OpenTreeOfLife[89]. We quantified the total heterogeneity[90] not due to sampling error as $I^2$, which can range from 0–100. We quantified partial heterogeneity explained by each random effect using the function i2_ml from the orchard package[91].

We created meta-regressions to investigate how moderators modulated the effects of advancing male age on ejaculate traits (aim 2). In all meta-regressions, we included the same random effects and phylogenetic matrix as in our null model and effect size (Zr) as our response variable. We first conducted a meta-regression with all moderators for which data were available for >75% of effect sizes and studies ("full" model). This full model was used to estimate the proportion of heterogeneity explained by moderators[92] while accounting for the confounding effects of other moderators. The full model included moderators of taxonomic class, ejaculate trait, proportion of maximum adult lifespan sampled, whether or not males had control over ejaculation, population type, sampling method of males, method of age estimation, whether or not a study was experimental, and whether or not males underwent "unnatural" manipulations. We then built several meta-regressions to explore individually the effects of each methodological and biological moderator (see Tables 1 and 2, Supplementary Notes 7, most of which had been pre-registered at OSF: https://osf.io/dk8sq/). Here, we also tested how the youngest and oldest ages sampled of the associated species (as a proportion of the maximum lifespan of the species) affected the evidence for senescence. We further tested the influence of the gonadosomatic index of species (GSI), which was not included in the full model, as it only had data for <75% of studies and was not pre-registered.

For each meta-regression model, we calculated the total heterogeneity ($Q_M$) and the proportion of total heterogeneity explained by moderators (marginal $R^2$), with the function r2_ml using the orchard package[91]. P values ($\alpha = 0.05$) indicate whether the heterogeneity explained was significant or not[90]. We created models without an intercept to test whether each level of a moderator showed evidence for senescence or improvement in ejaculate traits with age. However, for moderators with two levels, we were additionally interested in comparing effect sizes in one level to those in the other level. In such cases, we created a model with one level of the moderator as the intercept (here, a P value expressed whether one level of the moderator was different from the other level).

Taxonomic classes of Insecta, Actinopterygii, Aves, and Mammalia were over-represented classes in our dataset, each with >150 effect sizes from >30 studies (Supplementary Fig. 2). We thus created four separate meta-regressions for each class, with ejaculate trait as a moderator. Moreover, four species: lab mice (*Mus musculus*), lab rats (*Rattus norvegicus*), chicken/red junglefowl (*Gallus spp.*), and bulls (*Bos taurus*) were over-represented in our dataset (each species had >150 effect sizes across >20 studies; Supplementary Fig. 2). For these species, we created separate meta-regression models with ejaculate trait as a moderator.

Shapes of reproductive ageing are often curvilinear, characterised by an initial period of maturation, where performance increases from early- to mid-adult life and subsequently decreases (i.e. senescence) in late-adult life[4,53]. To test whether the effects of male age on ejaculate traits were curvilinear, we used linear mixed-effects models[93] (Supplementary Notes 11). These analyses were limited to traits which were measured on the same scale and units across studies/taxa.

We also used data from studies that measured age-dependent changes in both ejaculate traits and reproductive outcomes. Then, we ran a meta-regression using a type of trait (reproductive outcome or ejaculate trait) as a moderator (aim 3).

## Publication bias

We conducted a sensitivity analysis of our null model by replacing the random effects terms of cohort and effect size ID with a variance–covariance matrix[94]. We also performed various publication bias tests[94] (funnel plot, trim and fill multi-level meta-regression, and selection model; Supplementary Notes 12). These analyses were done to test for biased sampling of effect sizes in our study based on their precision, magnitude, publication year, and sample size.

## Other sensitivity analysis

We conducted two additional sensitivity analyses. First, we accounted for the low proportions of maximum adult lifespans sampled by

studies in our meta-analysis. Here, we re-ran our null model and models for the taxonomic classes of Insecta, Mammalia, Aves, and Actinopterygii, only using data from studies that sampled >10% of the maximum adult lifespan of the species. Second, we classified study aims as being explicitly interested in senescence or not. Studies that mentioned "ageing", "ageing", "senescence", "senescent", or "senescing" in their abstracts or titles were classified as explicitly interested in senescence. We then created a meta-regression with the study aim (i.e. interested in senescence or not) as our moderator to test whether studies that were interested in senescence showed senescence in ejaculate traits overall.

### Reporting summary

Further information on research design is available in the Nature Portfolio Reporting Summary linked to this article.

## Data availability

The data generated in this study have been deposited in the Open Science Framework database (https://osf.io/dk8sq/) with the following https://doi.org/10.17605/OSF.IO/DK8SQ. The data are available without any restricted access. The raw data are available under the file name "raw_data.csv". The processed data are available under the file name "spermFinalAllData.csv". The data used to produce the manuscript figures are provided in the Source Data file. Source data are provided in this paper.

## Code availability

All associated code can be found at the Open Science framework database (https://osf.io/dk8sq/) with the following https://doi.org/10.17605/OSF.IO/DK8SQ.

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

## Acknowledgements
We are extremely grateful to all the researchers who provided us with unpublished/missing data and life tables for various species, without whom our meta-analysis would not be possible: Abdallah Assiri, Adolfo Cordero, Adrienne Crosier, Alberto Velando, Alfonso Bolarin, Aline Malawey, Alistair Senior, Anders Pape Moller, Anil Kumar, Antje Girndt, Ashley Watt, Asim Orem, Bradley Metz, Bryan Neff, Budhan Pukazhenthi, Charles Fox, Chris Friesen, Chris Weldon, Christine Miller, Christophe Bressac, Claudia Fricke, Claudio Maia, Claus Wedekind, Clelia Gasparini, Clint McDonald, Craig Packer, Daniel Sasson, Davnah Payne, Diana Perez Staples Folger, Martha Reyez Hernandez, Distl Ottmar, Elena Zambrano, Emily "Becky" Cramer, Emily Duval, Erin MacCartney, Felipe Martinez, Fumio Hayashi, Gabriele Sorci, Gerard Wilkinson, Gerlind Lehmann, Gregor Majdic, Hasan Sevgili, Heriberto Martinez, Ilie Racotta, Ioannis Tsakmakidis, Jan Lifjeld, Jane Hurst, Janice Bailey, Maurice Clotilde, Jesus Dorado, Jurgen Heinze, Karen Lockyear, Karolina Stasiak, Katarzyna Kotarska, Kathrin Langen, Klaus Reinhardt, Leandro Miranda, Leslie Curren, Linda Penfold, Maira Brito, Malgorzata Kruczek, Manasi Kanuga, Marion Mehlis, Mark Elgar, Martin Brinkworth, Martin Schulze, Maud Bonato, Megan Head, Melissah Rowe, Michael Greenfield, Michael Ritchie, Michele Di Iorio, Michelle Helinski, Milos Krist, Moira O'Bryan, Muhammed Ines Inanc, Naomi Pierce, Nicolaia Iaffaldano, Nikos Papadopolous, Nikron Thongtip, Nils Cordes, Nucharin Songsasen, Megan Brown, Pablo Bermejo Alvares, Paco Garcia Gonzales, Panos Milonas, Patricia Diogo, Paul Joseph, Philip Downing, Priscilla Ramos, Rakesh Seth, Stuart Reynolds, Rebecca Dean, Sachiko Koyama, Satoshi Hiroyoshi, Silvia Cerolini, Sina Metzler, Stanislaw Kondrack, Stefan Luepold, Steven Ramm, Stuart Meyers, Theo Bakker, Tobias Kehl, Triin Hallap, Ulrike Luderer, Upama Aich, Wael Farag, Wei Shi, Wen Liao, Xiaoxu Li, Yasaman Alavi, Yih Fwu Lin, and Yingmei Zhang. We are also thankful to Milan Vrtilek, Rose O'Dea, Kevin Foster, Ana Silva, and Ellie Bath for their helpful comments and suggestions, as well as members of the Biology of Sperm Conference, 2023, for constructive criticism. Finally, we thank Neil Gemmell who supported preliminary meta-analytic investigations conducted by SJ and SN. KS was supported by an SSE Rosemary Grant award. R.V.T. and T.P. were supported by a BBSRC Standard Grant (BB/V001256/1). S.N. was supported by an Australian Research Council (ARC) Discovery Project Grant (DP210100812). S.J. was supported by a Royal Society of New Zealand Grant. R.S.G. was supported by a NERC Independent Research Fellowship (NE/M018458/1). I.S. was supported by a Biotechnology and Biological Sciences Research Council (BBSRC) Fellowship (BB/T008881/1), a Royal Society Dorothy Hodgkin Fellowship (DHF\R1\211084), and a Wellcome Institutional Strategic Support Fund, University of Oxford (BRR00060).

## Author contributions
K.S., R.V.T., T.P., and I.S. designed the study. K.S., S.G., and R.V.T. screened the studies. S.N. and S.J. provided extra studies. R.S.G. suggested methods to standardise age. K.S. extracted data. R.V.T. checked for repeatability of data extraction. R.V.T., S.N., S.J., and K.S. wrote the code and analysed the data. K.S., R.V.T., and I.S. wrote the first draft of the paper. All authors contributed in the critical assessment of the paper and subsequent revisions.

 **12**

## Competing interests

The authors declare no competing interests.
