## [Peer Review File · Nature Communications]

Meta-analysis shows no consistent evidence for senescence in ejaculate traits across animalsREVIEWER COMMENTS

Reviewer #1 (Remarks to the Author):

I found this to be an interesting and surely valuable analysis. Although inconclusive in some respects, it is an impressive dataset that generates many interesting new avenues for future research.

A few points I would like to see clarified:

Line 103: the inclusion criteria just required two different adult sampling points, so I'm a little concerned that, if reproductive performance peaks somewhere in "mid" adulthood, this potentially confounds some studies of males "on the way up" and others "in decline". To be fair, the authors themselves later recognise the problem in the discussion, but I wonder if more could be done to try and control for this? If already estimating the proportion of reproductive lifespan covered, could this info be combined with the starting point to try to "anchor" each of the different measurements more meaningfully? (Related to above: were quadratic effects more likely to be observed if a higher proportion of the reproductive lifespan was sampled?)

Line 178: I didn't understand what effect size ID accounts for here (only one effect size per trait/study combination?). Please clarify.

Line 242: I found it surprising that between species differences accounted for none of the variation in this analysis, and am uncertain how this should be interpreted. Can the authors expand on this point?

Line 289: doesn't taking a ratio assume that testis size scales isometrically? Would it perhaps be preferable to include testis and body size as covariates?

Line 406: not all insects have continuous sperm production though. I'm not sure how that maps to your dataset, but there might be the possibility to compare?

Minor corrections

Line 89: Bielefeld Academic?

Figure 2: a minor suggestion, but for ease of comparison: consider adjusting the spacing so that each of the traits occupies the same "row" (with gaps as appropriate if some traits are missing data for some taxa)

Reviewer #2 (Remarks to the Author):

In this study, the authors conducted a meta-analysis to quantify the impact of advancing male age on ejaculate traits across various non-human animal species. The central finding of this study is the lack of consistent evidence for ejaculate senescence across these species. However, when exploring the influence of specific individual moderators, the results indicate that age-related alterations in ejaculate traits do manifest in certain taxa/species, especially when there is a higher proportion of the maximum lifespan covered in the study. On the whole, I find this study intriguing, well-written and organized, although there are some methodological aspects that require clarification and certain sections that could benefit from enhancement. The comments and suggestions provided below are intended to assist the authors in elevating the quality of their manuscript.

1) One of my main points of concern pertains to the proportion of lifespan sampled (referred to as *Lssampled* in the database) in the studies incorporated into the analyses. In general, the studies included in the meta-analyses tend to span only a fraction of the adult lifespan of the species under investigation. For instance, the average proportion of lifespan covered by the studies within the database is approximately 25%, with a median value of around 20%. Given that the onset of senescence often occurs later in life, this underrepresentation of studies covering the majority of the adult reproductive lifespan might account for a) the authors' inability to detect a distinct overarching senescence pattern across animal species, and b) the increased likelihood of identifying senescence when a higher proportion of lifespan is studied. It is important for the authors to address this limitation prior to making definitive statements about the absence of consistent evidence for ejaculate senescence in animals. To this end, it would be valuable to demonstrate whether similar outcomes are obtained when studies covering a smaller proportion of the adult lifespan are excluded from the analyses (e.g., $\leq 50\%$). Additionally, I

encourage the authors to present the maximum and minimum ages at sampling, along with the age at maturity employed to estimate the proportion of lifespan sampled in the database, rather than just presenting the estimated proportion (L_{sampled}).

2) The analysis of age-related changes in ejaculate traits employed a linear age function.

However, this approach could be problematic due to the well-established non-linear nature of many reproductive traits, including those linked to sperm senescence and male reproductive success. While I lack big expertise in meta-analyses, I think that the metaphor package also allows for the incorporation of nonlinear relationships, such as polynomial and spline models. If feasible, I am curious why the authors did not explore these alternatives.

3) Lines 25-26: the explanation for the absence of senescence is confusing, so please clarify.

Moreover, it's worth noting that other explanations unrelated to indeterminate growth, such as the maintenance of cellular repair capacity throughout the lifespan or the extension of telomeres, could also contribute.

4) Lines 67-68: Different theories of ageing predict different outcomes related to sperm senescence. I would recommend the authors be more explicit or rephrase the sentence.

Furthermore, it's crucial to acknowledge the distinction between aging and senescence as separate processes.

5) Lines 153-158: without a more detailed rationale, it's unclear why parameters like egg numbers, egg viability, hatchability, or offspring body condition are considered suitable proxies for male reproductive outcomes. Indeed, these traits are often significantly influenced, and sometimes determined, by maternal rather than paternal phenotypes.

Moreover, when examining these traits, accounting for the presence or absence of parental care within the species and the potential impact of certain female traits (e.g., age category) could be beneficial.

6) Replace 'male reproductive senescence' with 'ejaculate senescence' in line 353.

7) Lines 431-444 would benefit from an expanded discussion of the various ways in which females could have influenced these findings. Addressing this aspect in more detail would enhance the manuscript's overall clarity.

8) Lastly, it would be advantageous to include a comprehensive discussion on how variations in post-meiotic sperm senescence and parental care might elucidate some of the observed results.

Reviewer #3 (Remarks to the Author):

The authors presented a meta-analysis on the effect of age on ejaculate traits across animals. No consistent effect of age was found across species, this result goes against the theoretical prediction that reproductive senescence should be pervasive across animals. The authors explained the heterogeneity of patterns found using different biological and methodological moderators. They notably found that the percentage of lifespan sampled influence positively the likelihood to find senescence and found some interesting differences between taxa and ejaculate traits studied. Overall, I enjoyed reading this manuscript as this study is both presenting the state of art regarding the evidence of male reproductive senescence but more importantly it highlights the knowledge gaps associated to this field and provide direction for future research. I think this study will make a valuable contribution to Nature Communication. The authors collected an impressive dataset for this analysis. The meta-analytic procedure was well performed and described so I have not much to say on the methods. My only main criticism would be regarding the inclusion of studies with very low age range in the analysis which I will detail more here in addition to other minor comments.

The authors rightly acknowledged the issue that some studies might not measured a sufficient age range of individuals to find any reproductive senescence by using the percentage of lifespan covered as a moderator. Those results are presented in figure 4. What really concerned me here was the amount of effect sizes with very low lifespan coverage. For instance, a large proportion of effect sizes have a lifespan coverage below 25% and there are even studies very close to 0%. With such a low age coverage I think it could be impossible to correctly assess the pattern of reproductive senescence. I was wondering whether keeping those effect sizes is really adding any meaningful information to the analysis. Would it be possible to do this analysis with some kind of minimum age coverage threshold? I understand it is not ideal because the authors would have to set an arbitrary threshold but at least studies with very low coverage would be removed.

I have spotted several issues and misleading sentences in Table 1:

- I did not understand why shape of ageing is included in this table as a biological moderator. It was never used as a moderator in the analysis. It was only accounted for in a different analysis with quadratic models. Shape is only a property of the senescence pattern and not a moderator.

- The section of life history strategies and mortality risks should be merged. The pace of life of individuals is the result of mortality risks so it does not make sense to present them as independent moderators.

- The cross sectional versus longitudinal sampling part is a bit misleading, the authors rightly stated that having longitudinal data could allow us to correct for selective disappearance. However, I do not think it is done at all in this study. My understanding is more that longitudinal data are categorized as such only if males were measured repeatedly and not if the analysis accounted for selective disappearance. In this specific meta-analysis, it is more longitudinal data analysed in a “cross-sectional way” from my understanding. Some changes to this section are needed to not mislead readers.

Line 135: Maximum lifespan data for each species is reported to calculate the percentage of lifespan coverage. Some animal species have quite strong sex-differences in lifespan, and it is not specified in the manuscript whether it was based only on male data or both sexes combined.

Line 178: What does the random effect “effect size ID” correspond to? It is not straightforward and should be explained in the manuscript.

In Figure 2, S5 and S6: Some overall effects are only based on 1 study (e.g. Figure 2B “Concentration”, “Motility”, “Morphology”, “Oxidative stress”). I’m quite sceptical about representing the means associated to those groups and even more about highlighting those results in the result section (line 274 for instance). It is interesting to still represent those effects sizes in the figure, but the authors might consider to not display the means based on low number of studies.

Figure 3: the use of the term “negative senescence” might be a bit too much here. Strong evidence are needed to find such pattern. An improvement can just be associated to the end of growth during the start of the adulthood and not to negative senescence through adulthood.

Lines 361-362: Many studies included in this analysis are not explicitly testing senescence. Is it possible to quantify the percentage of studies included in this meta-analysis that are explicitly testing senescence? In addition, testing quantitatively if studies interesting explicitly in senescence really have a higher lifespan coverage would be nice to set guidelines for future studies. It is a different thing to say that studies are simply not looking at something or that they are looking at it but using the wrong method.

Lasty I was a little surprised that humans were not mentioned at all in this manuscript. I understand that the authors did not want to include them in the meta-analysis but I would have liked as a reader few sentences in the discussion to compare the results of this analysis to what is generally found in humans. For instance, are the patterns of men senescence also depending on the trait studied? It also makes sense to compare this meta-analysis results to the human species in which the data quality could be consider very high if not the best.

REVIEWER COMMENTS

We are very grateful to all three reviewers for their constructive, positive, and insightful comments, which have greatly improved our manuscript. We have re-analysed our data in line with comments from all reviewers, which mainly focused on addressing how low proportion of lifespans sampled (LS), curvilinear ageing functions, and studies not explicitly testing for senescence, could affect our results. In light of the concerns about these issues raised by reviewers, we have emphasized that our study could have underestimated senescence, and provide suggestions on how to interpret our study considering these limitations (e.g. L 15-18, 345-347).

Note: changes made in response to reviewer comments are highlighted in our manuscript in orange

Reviewer #1 (Remarks to the Author):

I found this to be an interesting and surely valuable analysis. Although inconclusive in some respects, it is an impressive dataset that generates many interesting new avenues for future research.

Response: We thank the reviewer for their positive feedback. We clarify the points raised by the reviewer below.

A few points I would like to see clarified:

Comment 1. Line 103: the inclusion criteria just required two different adult sampling points, so I'm a little concerned that, if reproductive performance peaks somewhere in "mid" adulthood, this potentially confounds some studies of males "on the way up" and others "in decline". To be fair, the authors themselves later recognise the problem in the discussion, but I wonder if more could be done to try and control for this? If already estimating the proportion of reproductive lifespan covered, could this info be combined with the starting point to try to "anchor" each of the different measurements more meaningfully? (Related to above: were quadratic effects more likely to be observed if a higher proportion of the reproductive lifespan was sampled?)

Response 1: We agree with the reviewer that if ageing is a curvilinear function, and the starting/end points sampled are different for studies, then some studies will sample males on the "way up" and some on the "way down". Thus, different studies could be sampling the same percentage of lifespan (LS: say 40%) but at different time points/stages of ontogeny (e.g. from 10-50% of LS versus 50-90% of LS).

To address this issue, we have taken the following approaches:

1. We have constructed two additional meta-regression models that test how the minimum (youngest) and maximum (oldest) age sampled (as % of maximum LS) affects the evidence for senescence (L 155-159, 482-484, Fig. S9, S10, S11). These analyses should address the question of how the lowest age and highest age sampled might affect our results.

2. We have included in the Supplementary Information, a scatterplot that represents (for each study) the stage and range of ontogeny of a given species for where it has been sampled (i.e. the youngest age, oldest age, and % LS sampled; L 155-159, 482-484, Fig. S9). This should allow readers to infer at what stage animals in a study were sampled and for how long.

We would like to point out that the shape of reproductive ageing trajectories varies drastically between taxa (see Jones et al, 2014). Consequently, for some taxa, senescence might decline linearly and onset from an early age, while for others, senescence might only occur very late in life, and have a convex shape. Thus, sampling from early- to mid-life might still yield evidence for senescence for some species (as seen in Fig. 4A in our manuscript).

3. In our previous draft, we used LMMs to test for curvilinear effects of age, for traits that were on the same scale (0-100%) across species/studies. As proof of concept, we have now attempted to do the analysis that reviewer 1 suggests. Specifically, we have “anchored” values of the youngest age group in each effect size/row in a study, and then compared the ratio of change in trait values, of each subsequent age group to the youngest age group (we focus on sperm number in insects to exemplify this).

For instance, if the youngest group produced 500 sperm, the second age group produced 1500 sperm, and the third and fourth age groups produced 1000 and 200 sperm, respectively, these four age groups will get the values of 1, 3, 2, and 0.4, respectively. We only use studies where >2 age groups have been sampled to be able to model the effects as quadratic. Our plots show that these traits (e.g. sperm number) are on very different scales for different studies/taxa, making it difficult to interpret the effect of age using quadratic models (Figure R1A). In Figure R1B, we log₁₀ transform the Y axis to ease the interpretation of the figure, as values across studies are easier to visualise when they are on the same scale. While the overall effect of age on standardised sperm number when averaged across all studies is quadratic, individual studies do not consistently show a late-life decline in sperm number (Figure R1B).

We consider such an analysis of anchoring the youngest age group’s values to be inappropriate because it leads to effects that are incomparable across studies, due to the Y axis being on drastically different scales and measured in different units. Anchoring the youngest age group leads to effects of vastly different magnitudes between different studies (thus emphasizing the need of using effect sizes and meta-analysis), because anchoring the youngest age and comparing all subsequent ages to the youngest age, make the effect of age multiplicative. We have thus not extended such an analysis to other traits or presented this analysis in our manuscript. However, we hope that visually, reviewer 1 and the readers of the manuscript would be able to infer whether animals were sampled “on their way up” or “down”, and their specific sampling stage, from Figure R1 (presented below), and from the analysis on the youngest (minimum) and oldest (maximum) age of sampling (presented in the manuscript - Fig. S9, S10, S11).

Figure R1A: Insect sperm number (Y axis- standardised trait) standardised across various species and studies, by dividing values of sperm number in each age class in a study, by the value of sperm number in the youngest age class of that study. X axis standardised as the proportion of maximum adult lifespan represented by an age class where males were sampled. Each point is weighted by the log sample size of males in that age group in that study. The dark orange line shows average across all studies in the analysis, light lines show individual studies. Shaded area shows 95% CI. Only studies (N=46) with more than 2 age groups sampled were included in the analysis to allow modelling of quadratic patterns.

Figure R1B: Insect sperm number (Y axis) standardised across various species and studies (on log scale), by dividing values of sperm number in each age class in a study by the value of sperm number in the youngest age class of that study, and then log10 transforming the effect. X axis standardised as the proportion of maximum adult lifespan represented by an age class where males were sampled. Y axis is on

a log scale because on a linear scale (Figure R1A), the points are on very different units thus prevent any meaningful visual interpretation of the shape of ageing and stages at which individuals were sampled. Each point is weighted by the log sample size of males in that age group in that study. The dark orange line shows average across all studies in the analysis, light lines show individual studies. Shaded area shows 95% CI. Only studies (N=46) with more than 2 age groups sampled were included in the analysis to allow modelling of quadratic patterns.

Comment 2. Line 178: I didn't understand what effect size ID accounts for here (only one effect size per trait/study combination?). Please clarify.

Response 2: Effect size ID is a unit-level/effect size level random effect, and represents the residual within-study variance (i.e. variance unexplained by study/cohort within study/species/phylogeny) (Yang et al, 2023). We have now clarified this in the MS (L 457-458). In a normal model specification, we do not need to explicitly include such a random effect but the `rma.mv` function in `metafor` requires us to model this explicitly.

Comment 3: Line 242: I found it surprising that between species differences accounted for none of the variation in this analysis, and am uncertain how this should be interpreted. Can the authors expand on this point?

Response 3: Such a result of species explaining 0% variance and phylogeny explaining non-zero variance is not unusual. This result is because all the between-species variance is explained by phylogenetic relatedness. In the absence of phylogeny, species would explain non-zero variance. In our models, species variance should be interpreted as non-phylogenetic variance (i.e. species similarities due to factors other than phylogeny, such as ecology). See Cinar et al. (2022) for more details on this very point.

Comment 4: Line 289: doesn't taking a ratio assume that testis size scales isometrically? Would it perhaps be preferable to include testis and body size as covariates?

Response 4: Our aim was to test how the degree of sperm competition might affect evidence for senescence in ejaculate traits. Gonadosomatic index (GSI: ratio of testis mass to body mass) was used as a proxy for level of sperm competition. Given our aim was not to test how the size of testes itself affects senescence, we have not included testis and body size in our analyses. We now clarify this (L 144-145).

Comment 5: not all insects have continuous sperm production though. I'm not sure how that maps

Response 5: Even for species that do not show continuous spermatogenesis, but have very low rates of sperm loss/resorption/reabsorption, old virgin males might still produce a larger ejaculate than young virgin males. Thus, one would need data on both, sperm loss/reabsorption rates as well as spermatogenesis rates to infer how mating frequency and age could interact to affect sperm accumulation. Unfortunately, such data are unavailable for most taxa, and are biased toward model species like *Drosophila* spp. For example, in *Drosophila* spp., spermatogenesis is continuous/life-long (Demarco et al 2014; Santos, 2023). However, spermatogenesis rates reduce in late life (Sepil et al, 2020) but

males overall show very little sperm loss (Bjork et al, 2007), and can keep accumulating large numbers of sperm with age, when left unmated (Sepil et al, 2020). We would have added such data in our analysis if it was easily available for more species. We have however, worded our argument in a more nuanced way (L 274-278).

Comment 6: Minor corrections Line 89: Bielefeld Academic?

Response 6: Indeed, thank for you for spotting this.

Comment 7: Figure 2: a minor suggestion, but for ease of comparison: consider adjusting the spacing so that each of the traits occupies the same "row" (with gaps as appropriate if some traits are missing data for some taxa)

Response 7: The figure would look asymmetrical due to gaps being left at the bottom of taxa with fewer traits. We have thus left the manuscript figure unchanged.

Reviewer #2 (Remarks to the Author):

Comment 8: In this study, the authors conducted a meta-analysis to quantify the impact of advancing male age on ejaculate traits across various non-human animal species. The central finding of this study is the lack of consistent evidence for ejaculate senescence across these species. However, when exploring the influence of specific individual moderators, the results indicate that age-related alterations in ejaculate traits do manifest in certain taxa/species, especially when there is a higher proportion of the maximum lifespan covered in the study. On the whole, I find this study intriguing, well-written and organized, although there are some methodological aspects that require clarification and certain sections that could benefit from enhancement. The comments and suggestions provided below are intended to assist the authors in elevating the quality of their manuscript.

Response 8: We thank the reviewer for their constructive comments.

Comment 9: 1) One of my main points of concern pertains to the proportion of lifespan sampled (referred to as L_{sampled} in the database) in the studies incorporated into the analyses. In general, the studies included in the meta-analyses tend to span only a fraction of the adult lifespan of the species under investigation. For instance, the average proportion of lifespan covered by the studies within the database is approximately 25%, with a median value of around 20%. Given that the onset of senescence often occurs later in life, this underrepresentation of studies covering the majority of the adult reproductive lifespan might account for a) the authors' inability to detect a distinct overarching senescence pattern across animal species, and b) the increased likelihood of identifying senescence when a higher proportion of lifespan is studied. It is important for the authors to address this limitation prior to making definitive statements about the absence of consistent evidence for ejaculate senescence in animals. To this end, it would be valuable to demonstrate whether similar outcomes are obtained when studies covering a smaller proportion of the adult lifespan are excluded from the analyses (e.g., $\leq 50\%$).

Response 9: We endorse the point raised by reviewer 2 and have ourselves been aware of this issue (as previously highlighted in the manuscript). However, we now place more emphasis on this point in the new version of the manuscript.

We acknowledge the fact that many studies sample a lower proportion of lifespan. We have now re-analysed our data using only studies that sample >10% of LS (L 104-106, 184-188, 524-527). We present null and taxonomic class specific models (in Supplementary Information and sensitivity analyses). We chose >10% as an arbitrary cut-off because a higher cut-off (e.g. 20% or 50%) would have meant excluding too many studies and reducing statistical power for meaningful interpretation, especially when analysing the effects of biological moderators (i.e. trait * taxonomic class). This is because 279 out of 379 studies sampled >10% LS, 214 studies sampled >20%, and only 47 studies sampled >50% LS.

Results from this analysis where studies sampling <10% of maximum LS are excluded, show that there still is no significant effect of advancing age on ejaculate traits, and that our original results remain qualitatively unchanged (L 184-188; Fig. S20, S21).

We would like to point out three caveats with respect to our data on %LS sampled:

First, the LS estimated for many vertebrates come from large databases such as AnAge or Pantheria, and for most invertebrates, from other peer-reviewed studies, rather than from the same study in our meta-analysis (sources of LS can be found in supplementary data on OSF). Thus, these LS estimates may not always accurately reflect the maximum lifespans of the studied populations included in our meta-analysis (L 414-418).

Second, the onset of senescence for animals differs vastly (Jones et al, 2014). Thus, for some taxa, even a low % of LS sampled might be sufficient to be able to detect senescence. An example of this pattern in our study is *Gallus* spp. (chicken and red junglefowl). Here, senescence was detected in ejaculate size and sperm number, despite studies sampling a median of only 10% of the species' maximum LS. This result was possibly due to chicken breeds such as broilers being artificially selected for short generation times and fast life histories, thus possibly an early onset of senescence.

Third, while % LS sampled explained significant variance overall, this was mostly driven by LS sampled explaining significant variance for lab and captive animals (L 297). For domestic and wild animals, LS sampled did not explain significant variance, and it might not be that important a predictor for detecting senescence in these populations. In domestic animals, this result could be due to low reproductive quality males being culled by farmers as males age. This culling could result in old males with high reproductive quality surviving, thus representing a biased sample for older age categories. In wild animals, this could be driven by selective disappearance and condition-dependent mortality risk with age, whereby only high quality/condition males live to an older age.

We present the new analyses in our revised version of the manuscript. However, given the caveats of the data for %LS sampled, we have avoided over-interpreting the importance of %LS sampled.

Comment 10: Additionally, I encourage the authors to present the maximum and minimum ages at sampling, along with the age at maturity employed to estimate the proportion of lifespan sampled in the database, rather than just presenting the estimated proportion (L_{sampled}).

Response 10: We now present results for how the youngest (minimum) and oldest (maximum) age at sampling (as % of maximum lifespan) affects the results (L 104-106, 184-188, 524-527). We also provide the range of youngest and oldest ages sampled for each study (as a scatterplot in Supplementary Information), which should aid readers in assessing the stage of ontogeny of an animal as sampled by the study (Fig. S9 in manuscript; also see response 1 above).

Comment 11: 2) The analysis of age-related changes in ejaculate traits employed a linear age function. However, this approach could be problematic due to the well-established non-linear nature of many reproductive traits, including those linked to sperm senescence and male reproductive success. While I lack big expertise in meta-analyses, I think that the metafor package also allows for the incorporation of nonlinear relationships, such as polynomial and spline models. If feasible, I am curious why the authors did not explore these alternatives.

Response 11: We had certainly considered modelling effects of age as curvilinear in our meta-analysis, however this was unfeasible because of the following reasons:

1. The metafor package allows modelling of *non-linear effects of moderators*. For example, if we wished to test the effects of %LS sampled on effect sizes, we could do this using a polynomial function. However, in the metafor package, modelling non-linear moderators can only be done for models without random effects (this uses the `rma` function, while random effect models use the `rma.mv` function). Given the importance of including random effects in eco-evo meta-analyses (Nakagawa et al, 2023), we included study, cohort, and species identity, as well as phylogeny as random effects in our analyses, and thus were unable to do a non-linear analysis of moderators (and encourage meta-analysts to develop methods for such analyses).

2. Regarding modelling non-linear effect sizes, the metafor package does not allow modelling a *non-linear effect size*, and we are not aware of any method that even allows the calculation of non-linear effect sizes. The difficulty in calculating non-linear effect sizes is that it needs to combine standardised linear effects along with standardised quadratic effects, which has not been done for meta-analysis yet (personal communication: Shinichi Nakagawa). Thus, using just meta-analytical techniques, we were unable to measure curvilinear effects of age in a standardised and comparable way (i.e. as an effect size).

3. There are other reasons, not specific to meta-analysis/metafor, as to why a quadratic effect model (as our main analysis) would not be appropriate. A quadratic effect analysis would require at least three age groups to be measured by studies. Only 193 out of 379 studies in our meta-analysis reported data for three age groups or more, which would exclude about half our dataset (and possibly lead to biases).

The closest alternative we could think of to test for such non-linear effects in our dataset were LMMs, for traits where the Y axis could be standardised across studies and taxa in a meaningful way, to represent the magnitude of observed effects. We presented these

analyses in our original manuscript (L 146-148), for traits that can be easily standardised (i.e. traits measured as a proportion/percentage). As proof of concept for non-standardisable traits (e.g. insect sperm number), we have conducted a new analysis (see comment 1 above).

Despite these caveats, we do emphasise in the revised version of the manuscript that our linear analysis (using Zr as our effect size) might have underestimated senescence due to ageing function often being curvilinear (L 15-17, 345-347).

Comment 12: 3) Lines 25-26: the explanation for the absence of senescence is confusing, so please clarify. Moreover, it's worth noting that other explanations unrelated to indeterminate growth, such as the maintenance of cellular repair capacity throughout the lifespan or the extension of telomeres, could also contribute.

Response 12: We have now clarified this (L 43-49).

Comment 13: 4) Lines 67-68: Different theories of ageing predict different outcomes related to sperm senescence. I would recommend the authors be more explicit or rephrase the sentence. Furthermore, it's crucial to acknowledge the distinction between aging and senescence as separate processes.

Response 13: We have now made the distinction between ageing and senescence clear (L 80-82). With regards to theories of ageing specific to ejaculate traits, we think this was addressed in Table 1 (under moderators of ejaculate traits, degree of sperm competition), but is now explicitly mentioned in our predictions too (L 83).

Comment 14: 5) Lines 153-158: without a more detailed rationale, it's unclear why parameters like egg numbers, egg viability, hatchability, or offspring body condition are considered suitable proxies for male reproductive outcomes. Indeed, these traits are often significantly influenced, and sometimes determined, by maternal rather than paternal phenotypes.

Response 14: We agree that reproductive outcome traits might be influenced more by maternal rather than paternal condition. However, without a formal test for this hypothesis, such a claim cannot be made. We thus tested whether male age affects such reproductive outcomes and whether effects of age on ejaculates differ from those on reproductive outcomes. Our results suggest that male advancing age might not impact reproductive outcomes as much as ejaculates, therefore allowing us to attribute such effects to female-driven effects (as you rightly say). However, the discussion about how maternal versus paternal phenotypes drive male reproductive success goes beyond the scope of this study, and so we have not made any changes to this section.

Comment 15: Moreover, when examining these traits, accounting for the presence or absence of parental care within the species and the potential impact of certain female traits (e.g., age category) could be beneficial.

Response 15: Testing for female traits/age would have been ideal but there are reasons why we didn't do such analyses.

First, such an analysis would lead to more complex models (such as male age and female age interacting) without us having specific predictions for what the effects of such interactions should be. Second, studies in our meta-analysis rarely manipulated female age (data on when female ages were manipulated is available in raw_data.csv data sheet at OSF). Moreover, studies usually only used young females, thus preventing such a comparison to be done with enough statistical power. Additionally, such an analysis would require standardising female ages to % of their maximum LS sampled (like we did for males), and collecting female specific LS data for 157 species, which was not the aim of this study. We appreciate that it would be an interesting avenue for the future, but have not conducted any further analyses to address this point as the data is not available. Regarding parental care, only a few studies that measured reproductive outcomes actually measured the condition (e.g. body size) of offspring (which is likely affected by parental care). Other, more commonly measured reproductive outcomes (e.g. fertilisation success, number of offspring) are unlikely to be influenced by parental care. Thus, we have not conducted an analysis to test how parental care might affect reproductive outcomes. However, in Table 1, we have included parental care as a potential moderator whose influence future studies could test (highlighted). We have also emphasized the role of maternal investment (L 312-317) and the importance of measuring offspring traits (L 343-344).

Comment 16: 6) Replace male reproductive senescence with 'ejaculate senescence' in line 353.

Response 16: Now reworded

Comment 17: 7) Lines 431-444 would benefit from an expanded discussion of the various ways in which females could have influenced these findings. Addressing this aspect in more detail would enhance the manuscript's overall clarity.

Response 17: We have now included in our discussion how female driven effects could alleviate deleterious effects of old male ejaculates (L 312-317).

Comment 18: 8) Lastly, it would be advantageous to include a comprehensive discussion on how variations in post-meiotic sperm senescence and parental care might elucidate some of the observed results.

Response 18: A key prediction for how parental care might affect reproductive senescence is that it could exaggerate senescence in caring parents, if parents evolve to allocate energy toward care and offspring quality at the cost of lower energy allocated to future reproduction. We have now added parental care as a potential moderator of reproductive senescence (Table 1), whose influence future studies could test. We have now also added an extended discussion of post meiotic sperm storage (Table 1).

Due to lack of available data on either parental care or sperm storage for the species included in our study, we are unable to discuss their implications on our results any

further than suggesting them as potential moderators for future studies to test (now addressed in Table 1).

Reviewer #3 (Remarks to the Author):

Comment 19: The authors presented a meta-analysis on the effect of age on ejaculate traits across animals. No consistent effect of age was found across species, this result goes against the theoretical prediction that reproductive senescence should be pervasive across animals. The authors explained the heterogeneity of patterns found using different biological and methodological moderators. They notably found that the percentage of lifespan sampled influence positively the likelihood to find senescence and found some interesting differences between taxa and ejaculate traits studied. Overall, I enjoyed reading this manuscript as this study is both presenting the state of art regarding the evidence of male reproductive senescence but more importantly it highlights the knowledge gaps associated to this field and provide direction for future research. I think this study will make a valuable contribution to Nature Communication. The authors collected an impressive dataset for this analysis. The meta-analytic procedure was well performed and described so I have not much to say on the methods.

Response 19: We thank the reviewer for their kind words.

Comment 20: My only main criticism would be regarding the inclusion of studies with very low age range in the analysis which I will detail more here in addition to other minor comments. The authors rightly acknowledged the issue that some studies might not measured a sufficient age range of individuals to find any reproductive senescence by using the percentage of lifespan covered as a moderator. Those results are presented in figure 4. What really concerned me here was the amount of effect sizes with very low lifespan coverage. For instance, a large proportion of effect sizes have a lifespan coverage below 25% and there are even studies very close to 0%. With such a low age coverage I think it could be impossible to correctly assess the pattern of reproductive senescence. I was wondering whether keeping those effect sizes is really adding any meaningful information to the analysis. Would it be possible to do this analysis with some kind of minimum age coverage threshold? I understand it is not ideal because the authors would have to set an arbitrary threshold but at least studies with very low coverage would be removed.

Response 20: In line with your (and reviewer 2's) suggestions, we have re-analysed our data in two different ways to further test how % LS sampled could be impacting/biasing our results, and account for effects of % LS sampled. This has now been described above (see response to comments 9 and 10).

Comment 21: I have spotted several issues and misleading sentences in Table 1:
- I did not understand why shape of ageing is included in this table as a biological moderator. It was never used as a moderator in the analysis. It was only accounted for in a different analysis with quadratic models. Shape is only a property of the senescence pattern and not a moderator.

Response 21: You are correct, we have now deleted this.

Comment 22: - The section of life history strategies and mortality risks should be merged. The pace of life of individuals is the result of mortality risks so it does not make sense to present them as independent moderators.

Response 22: We have now changed this (Table 1).

Comment 23: - The cross sectional versus longitudinal sampling part is a bit misleading, the authors rightly stated that having longitudinal data could allow us to correct for selective disappearance. However, I do not think it is done at all in this study. My understanding is more that longitudinal data are categorized as such only if males were measured repeatedly and not if the analysis accounted for selective disappearance. In this specific meta-analysis, it is more longitudinal data analysed in a “cross-sectional way” from my understanding. Some changes to this section are needed to not mislead readers.

Response 23: Even when only means are compared, longitudinal studies can account for selective disappearance as long as all males are measured across all age groups. Alternatively, longitudinal studies can still account for selective disappearance if only males who are measured when old are included in the calculation of means for all age groups. In these two scenarios, an increase in means of older age groups would reflect *true* increases in average individual-level reproductive output, rather than be due to selective disappearance. Consequently, in such cases, longitudinal studies can control for selective disappearances even when only means of age groups are compared (without the need for individual-level data), thus being much more informative about patterns of ageing than cross-sectional studies.

However, when fewer males are sampled at older ages than young ages, individual-level data rather than just means of age groups are needed (even for longitudinal studies) to account for selective disappearance.

We have now clarified this and put forth a more nuanced argument (L 243-248).

Comment 24: Line 135: Maximum lifespan data for each species is reported to calculate the percentage of lifespan coverage. Some animal species have quite strong sex-differences in lifespan, and it is not specified in the manuscript whether it was based only on male data or both sexes combined.

Response 24: When collecting data for the lifespan sampled, we always prioritised collecting data on male-specific lifespans from the study, if provided. We further emailed authors from papers where male-specific lifespans for the associated species were not available in datasets/databases, to try to collect data for lifespans of males in their populations. When male-specific data was not available from any sources we looked at, we collected species-specific lifespans. We never collected lifespan data from a source when the source reported the sex of the measured individual(s) as female.

However, some of our data (especially for vertebrates) on maximum LS comes from large datasets/databases such as AnAge (De Magalhaes and Costa, 2009), Moller (2006), Carey and Judge (2008), which often did not report the sex of the measured individual(s) (i.e. reported on a species-specific level), and in these cases the maximum LS estimates could

be female-specific. We have made our best effort to collect as much data as possible but acknowledge this is a caveat of those large databases.

We have now clarified this in the methods (L 410-418) and in Supplementary Information (Supplementary section 6), and added that LS estimates in our analysis may not always accurately reflect the LS of the population studied in our meta-analysis due to these reasons.

All lifespan data and the sources of LS data can be found in the supplementary excel sheet (Life tables_GSI_mating_status.csv) and has previously been uploaded on OSF (https://osf.io/dk8sq/?view_only=510c32be19964ec49cdc52996a51d43a).

Comment 25: Line 178: What does the random effect “effect size ID” correspond to? It is not straightforward and should be explained in the manuscript.

Response 25: Effect size ID is a unit-level/effect size level random effect, and represents the residual within-study variance (i.e. variance unexplained by study/cohort within study/species/phylogeny) (Yang et al, 2023). We have now clarified this in the MS (L 457-458). See also comment 2 above.

Comment 26: In Figure 2, S5 and S6: Some overall effects are only based on 1 study (e.g. Figure2B “Concentration”, “Motility”, “Morphology”, “Oxidative stress”). I’m quite sceptical about representing the means associated to those groups and even more about highlighting those results in the result section (line 274 for instance). It is interesting to still represent those effects sizes in the figure, but the authors might consider to no display the means based on low number of studies.

Response 26: We have removed statements highlighting or interpreting results based on few studies (e.g. L 131, 134, 142).

Comment 27: Figure 3: the use of the term “negative senescence” might be a bit too much here. Strong evidence are needed to find such pattern. An improvement can just be associated to the end of growth during the start of the adulthood and not to negative senescence through adulthood.

Response 27: This is now changed.

Comment 28: Lines 361-362: Many studies included in this analysis are not explicitly testing senescence. Is it possible to quantify the percentage of studies included in this meta-analysis that are explicitly testing senescence? In addition, testing quantitatively if studies interesting explicitly in senescence really have a higher lifespan coverage would be nice to set guidelines for future studies. It is a different thing to say that studies are simply not looking at something or that they are looking at it but using the wrong method.

Response 28: Whether or not studies explicitly test for senescence can be challenging to assess because studies often do not clearly state their aims, have multiple aims (e.g.

Awrahaman et al, 2014; Burris and Dam, 2015; Rebar and Greenfield, 2017), or might “seem” like they are testing for senescence/aging (i.e. sample males to a large % of their LS and have multiple age groups), but do not explicitly aim to test for senescence (e.g. Kipper et al, 2017; Rollings et al, 2020).

However, we agree with reviewer 3’s point that if studies are ‘explicitly’ interested in aging/senescence, they might reveal stronger support for senescence than studies that “just happen” to measure ejaculate traits for different ages of males for various other reasons. We thus have attempted to classify whether studies in our meta-analysis were ‘explicitly’ interested in studying senescence or not. Then, only using studies that were explicitly interested in testing for senescence, we have re-run the analysis for the null model and the taxonomic class * trait models, as these are the most informative ones based on our results. These are now included in the sensitivity analysis section and Supplementary Information. The file with study titles and their classification has been provided on OSF (file name: “k. Aims.csv”). Note that the classification of studies was done blind to the effect sizes and results section of the study, to prevent any biases.

To *objectively* carry out this classification, any study that used the exact words “ageing”, “aging”, “senescence”, “senescent”, or “senescing”, in their abstracts or titles, was classified as one that was interested in [testing for] senescence.

Lifespan sampled between studies that tested for senescence in our objective classification (101 studies used the words aging, ageing, senescence, senescing, or senescent in their abstracts or titles), versus those that did not (273 studies), were 34% and 20% (means), respectively (Figure R2). However, even when only analysing the 101 studies that were interested in senescence (from our objective classification), we did not find a significant effect of age overall (i.e. no overall evidence for senescence), on ejaculates (Figure R3). Visually, most effect sizes and the overall mean were shifted to the left (i.e. evidence of senescence) of zero, but the C.I. were still overlapping zero, thus the overall effect was not significant (Figure R3). However, study aim (i.e. objectively interested or not interested in senescence) as a moderator explained a significant amount of heterogeneity in our data ($R^2 = 5.08\%$, $P < 0.001$).

We further analysed Insects, Mammals, Birds, and Fish from those 101 studies interested in senescence (Figure R4), in four separate models. However, due to the low number of studies in each class-specific ejaculate trait, this analysis did not have statistical power for any meaningful conclusions, thus this class*trait analysis has not been included in our meta-analysis and only presented below for the reviewer’s curiosity.

Figure R2: Proportion of maximum adult lifespan sampled by a study (Y axis) of the associated species, differed between studies that were objectively classified as interested in senescence (1) and studies that were classified as not interested in senescence (0). Means and SE shown.

Figure R3: Even when only studies that were objectively classified as interested in senescence (“Yes”) were included, we did not find significant evidence for senescence in ejaculate traits overall. The meta-analytical mean however, was more shifted towards showing senescence in studies that tested for senescence explicitly, compared to studies that did not test for senescence (“No”).

Figure R4: Taxonomic class-specific models for effects of age on ejaculate traits, using only the 101 studies objectively classified as interested in senescence. Most traits in each taxonomic class had very few studies thus low statistical power to infer anything meaningful. A: Insects, B: Fish, C: Birds, D: Mammals. The size of each data point represents the precision of the effect size (1/SE). The X axis represents values of effect sizes as Fisher's z-transformed correlation coefficient (Zr), while the Y axis shows the density distribution of effect sizes. The position of the overall effect is shown by the dark circle, with negative values depicting senescence in ejaculate traits and positive values showing improvement in ejaculate traits with advancing male age. Bold error bars (95% C.I) show whether overall effect size is significantly different from zero (*i.e.* not overlapping zero), while light error bars show the 95% P.I. of effect sizes. Sample sizes reported as: k = number of effect sizes (in brackets: number of studies).

The words “ageing”, “aging”, “senescence”, “senescent” or “senescing”, in our objective classification may not accurately represent the aims of the study, thus we also *subjectively* classified study aims as testing for senescence or not, based on their abstracts and titles. In addition to the 101 studies classified objectively as interested in senescence, another 51 studies were subjectively classified as interested in senescence. Repeatability between our objective and subjective classifications were fair ($R^2 = 0.55$; Cohen's kappa=0.708). Based on our classification of study aims, studies that were not explicitly interested in

senescence had primarily the following aims: 1.) Many studies (often on domestic animals) seemed instead interested in *describing* how reproductive function and physiology change in animals with various environmental and biological factors (including age) rather than test explicitly for senescence (e.g. Aurich et al, 2003; Bhavé et al, 2020; Goericke-Pesch et al, 2013; N= 135 studies). 2.) Others were interested in sexual selection-related questions, and often only had age as a covariate along with other factors, to explain variance in ejaculates (e.g. Garcia-Gonzalez and Simmons, 2005; Sardell and duVal, 2014; Wedekind et al, 2007; N= 37 studies). 3.) Others yet, were interested in testing the effects of specific treatments/environments on age-dependent reproduction (e.g. Perez-Staples et al, 2008; Roth et al, 2008; Taylor et al, 2019; N= 56 studies). 4.) Few studies were interested in testing methods, such as different sperm collection or preservation techniques, and used males of different ages (e.g. Tabarez et al, 2017; N= 11 studies). 5.) Lastly, some studies (especially in lab rodents), were interested in testing for effects of specific genes or transgenic lines on reproduction, and used males of different ages (e.g. Nakamura et al, 2010; Xu et al, 2014; N= 18 studies).

In a separate analysis, we re-analysed our data using studies that were *either* objectively *or* subjectively classified as interested in senescence (152 studies total). This second analysis was done so as to not miss relevant studies interested in senescence that did not use the words “ageing”, “aging”, “senescence”, “senescent” or “senescing” in their titles and abstracts (e.g. Gasparini et al, 2010; Jones et al, 2007; Kehl et al, 2013; Metzler et al, 2018; Pennig and Wrigley, 2018). This analysis again did not show a significant effect of age on ejaculates (Figure R5), and results from this analysis were similar to results from the objective studies-only analysis. In our revised version of the manuscript, we only present results from our analysis using objective classification of study aims.

Figure R5: Even when only studies that were either objectively or subjectively classified as interested in senescence were included, we did not find significant evidence for senescence in ejaculate traits overall. These results were qualitatively similar to results from the model that only included objective classifications, thus this figure and analysis has not been included in the MS.

We understand that some readers might be opposed to us including studies not interested in senescence, in our meta-analysis (different meta-analysts getting very different results for the same question is a big caveat in meta-analysis: Gould et al, 2023). However, we think that even studies that do not aim to test for senescence are still informative about whether organisms senesce or not, and can help us understand what the patterns of ejaculate ageing are for animals. It is also possible that studies “alter” their aims/hypotheses to ‘testing for senescence’ (i.e. HARKing), or are more likely to use the keywords mentioned above if they detect senescence, and thus such classification might be biased towards selecting studies that find senescence post-hoc. Lastly, the use of “senescence”- like words in study aims might be field-specific and might lead to an exclusion of relevant studies from fields that do not use such words. Due to these reasons, the difficulty in assessing study aims, and the lack of overall senescence despite reanalysis based on study aims, we do not think that results from our re-analysis should be over-interpreted. We have thus only presented these new analyses only as a sensitivity analysis. We hope that our transparency about such issues would allow readers to evaluate for themselves the extent of senescence in animals.

These analyses, their associated descriptions and discussions, can be found in our revised version of the manuscript (L 104-106, 189-197, 228-232, 527-532, Fig. S22, S23). Our classification of study aims has been uploaded to OSF (https://osf.io/dk8sq/?view_only=).

Comment 29: Lasty I was a little surprised that humans were not mentioned at all in this manuscript. I understand that the authors did not want to include them in the meta-analysis but I would have liked as a reader few sentences in the discussion to compare the results of this analysis to what is generally found in humans. For instance, are the patterns of men senescence also depending on the trait studied? It also makes sense to compare this meta-analysis results to the human species in which the data quality could be consider very high if not the best.

Response 29: We have now compared our meta-analysis results to one that tested effects of male age on ejaculates in humans (Johnson et al, 2015) (L 213-221).

References:

Aurich, C., Achmann, R. and Aurich, J.E., 2003. Semen parameters and level of microsatellite heterozygosity in Noriker draught horse stallions. *Theriogenology*, 60(2), pp.371-378.

Awrahaman, Z.A., Champion de Crespigny, F. and Wedell, N., 2014. The impact of W olbachia, male age and mating history on cytoplasmic incompatibility and sperm transfer in *Drosophila simulans*. *Journal of Evolutionary Biology*, 27(1), pp.1-10.

Bhave, K.G., Jawahar, K.T.P., Kumarasamy, P., Sivakumar, T., Joseph, C., Shirsath, T., Deshmukh, P. and Venkataramanan, R., 2020. Genetic and non-genetic factors affecting semen production and quality characteristics of Gir cattle breed under semi-arid climate. *Veterinary World*, 13(8), p.1714.

Bjork, A., Dallai, R. and Pitnick, S., 2007. Adaptive modulation of sperm production rate in *Drosophila bifurca*, a species with giant sperm. *Biology letters*, 3(5), pp.517-519.

Burris, Z.P. and Dam, H.G., 2015. Spermatophore production as a function of food abundance and age in the calanoid copepods, *Acartia tonsa* and *Acartia hudsonica*. *Marine Biology*, 162, pp.841-853.

- Carey, J.R. and Judge, D.S., 2002. Longevity records: Life spans of mammals, birds, amphibians, reptiles, and fish. *Monographs on population aging*, (8).
- Cinar, O., Nakagawa, S. and Viechtbauer, W., 2022. Phylogenetic multilevel meta-analysis: A simulation study on the importance of modelling the phylogeny. *Methods in Ecology and Evolution*, 13(2), pp.383-395.
- De Magalhães, J.P. and Costa, A.J., 2009. A database of vertebrate longevity records and their relation to other life-history traits. *Journal of evolutionary biology*, 22(8), pp.1770-1774.
- Demarco, R.S., Eikenes, Å.H., Haglund, K. and Jones, D.L., 2014. Investigating spermatogenesis in *Drosophila melanogaster*. *Methods*, 68(1), pp.218-227.
- García-González, F. and Simmons, L.W., 2005. Sperm viability matters in insect sperm competition. *Current biology*, 15(3), pp.271-275.
- Gasparini, C., Marino, I.A.M., Boschetto, C. and Pilastro, A., 2010. Effect of male age on sperm traits and sperm competition success in the guppy (*Poecilia reticulata*). *Journal of Evolutionary Biology*, 23(1), pp.124-135.
- Goericke-Pesch, S. and Failing, K., 2013. Retrospective analysis of canine semen evaluations with special emphasis on the use of the hypoosmotic swelling (HOS) test and acrosomal evaluation using Spermac®. *Reproduction in Domestic Animals*, 48(2), pp.213-217.
- Gould, E., Fraser, H.S., Parker, T.H., Nakagawa, S., Griffith, S.C., Vesk, P.A., Fidler, F., Hamilton, D.G., Abbey-Lee, R.N., Abbott, J.K. and Aguirre, L.A., 2023. Same data, different analysts: variation in effect sizes due to analytical decisions in ecology and evolutionary biology.
- Johnson, S.L., Dunleavy, J., Gemmell, N.J. and Nakagawa, S., 2015. Consistent age-dependent declines in human semen quality: a systematic review and meta-analysis. *Ageing research reviews*, 19, pp.22-33.
- Jones, T.M., Featherston, R., Paris, D.B. and Elgar, M.A., 2007. Age-related sperm transfer and sperm competitive ability in the male hide beetle. *Behavioral Ecology*, 18(1), pp.251-258.
- Jones, O.R., Scheuerlein, A., Salguero-Gómez, R., Camarda, C.G., Schaible, R., Casper, B.B., Dahlgren, J.P., Ehrlén, J., García, M.B., Menges, E.S. and Quintana-Ascencio, P.F., 2014. Diversity of ageing across the tree of life. *Nature*, 505(7482), pp.169-173.
- Kehl, T., Karl, I. and Fischer, K., 2013. Old-male paternity advantage is a function of accumulating sperm and last-male precedence in a butterfly. *Molecular ecology*, 22(16), pp.4289-4297.
- Kipper, B.H., Trevizan, J.T., Carreira, J.T., Carvalho, I.R., Mingoti, G.Z., Beletti, M.E., Perri, S.H.V., Franciscato, D.A., Pierucci, J.C. and Koivisto, M.B., 2017. Sperm morphometry and chromatin condensation in Nelore bulls of different ages and their effects on IVF. *Theriogenology*, 87, pp.154-160.
- Metzler, S., Schrempf, A. and Heinze, J., 2018. Individual-and ejaculate-specific sperm traits in ant males. *Journal of insect physiology*, 107, pp.284-290.
- Møller, A.P., 2006. Sociality, age at first reproduction and senescence: comparative analyses of birds. *Journal of Evolutionary Biology*, 19(3), pp.682-689.
- Nakagawa, S., Yang, Y., Macartney, E.L., Spake, R. and Lagisz, M., 2023. Quantitative evidence synthesis: a practical guide on meta-analysis, meta-regression, and publication bias tests for environmental sciences. *Environmental Evidence*, 12(1), pp.1-19.
- Nakamura, B.N., Lawson, G., Chan, J.Y., Banuelos, J., Cortés, M.M., Hoang, Y.D., Ortiz, L., Rau, B.A. and Luderer, U., 2010. Knockout of the transcription factor NRF2 disrupts spermatogenesis in an age-dependent manner. *Free Radical Biology and Medicine*, 49(9), pp.1368-1379.
- Penning, K.A. and Wrigley, D.M., 2018. Aged *Eisenia fetida* earthworms exhibit decreased reproductive success. *Invertebrate reproduction & development*, 62(2), pp.67-73.
- Perez-Staples, D., Harmer, A.M., Collins, S.R. and Taylor, P.W., 2008. Potential for pre-release diet supplements to increase the sexual performance and longevity of male Queensland fruit flies. *Agricultural and forest entomology*, 10(3), pp.255-262.
- Rebar, D. and Greenfield, M.D., 2017. When do acoustic cues matter? Perceived competition and reproductive plasticity over lifespan in a bushcricket. *Animal Behaviour*, 128, pp.41-49.

- Rollings, N., Waye, H.L., Krohmer, R.W., Uhrig, E.J., Mason, R.T., Olsson, M., Whittington, C.M. and Friesen, C.R., 2020. Sperm telomere length correlates with blood telomeres and body size in red-sided garter snakes, *Thamnophis sirtalis parietalis*. *Journal of Zoology*, 312(1), pp.21-31.
- Roth, O., Ebert, D., Vizoso, D.B., Bieger, A. and Lass, S., 2008. Male-biased sex-ratio distortion caused by *Octosporea bayeri*, a vertically and horizontally-transmitted parasite of *Daphnia magna*. *International Journal for Parasitology*, 38(8-9), pp.969-979.
- Sardell, R.J. and DuVal, E.H., 2014. Small and variable sperm sizes suggest low sperm competition despite multiple paternity in a lekking suboscine bird. *The Auk: Ornithological Advances*, 131(4), pp.660-671.
- Santos, I.B., Wainman, A., Garrido-Maraver, J., Pires, V., Riparbelli, M.G., Kovács, L., Callaini, G., Glover, D.M. and Tavares, Á.A., 2023. Mob4 is essential for spermatogenesis in *Drosophila melanogaster*. *Genetics*, 224(4), p.iyad104.
- Sepil, I., Hopkins, B.R., Dean, R., Bath, E., Friedman, S., Swanson, B., Ostridge, H.J., Harper, L., Buehner, N.A., Wolfner, M.F. and Konietzny, R., 2020. Male reproductive aging arises via multifaceted mating-dependent sperm and seminal proteome declines, but is postponable in *Drosophila*. *Proceedings of the National Academy of Sciences*, 117(29), pp.17094-17103.
- Tabarez, A., García, W. and Palomo, M.J., 2017. Effect of the type of egg yolk, removal of seminal plasma and donor age on buck sperm cryopreservation. *Small Ruminant Research*, 149, pp.91-98.
- Taylor, J.D., Baumgartner, A., Schmid, T.E. and Brinkworth, M.H., 2019. Responses to genotoxicity in mouse testicular germ cells and epididymal spermatozoa are affected by increased age. *Toxicology Letters*, 310, pp.1-6.
- Wedekind, C., Rudolfson, G., Jacob, A., Urbach, D. and Müller, R., 2007. The genetic consequences of hatchery-induced sperm competition in a salmonid. *Biological Conservation*, 137(2), pp.180-188.
- Xu, B., Washington, A.M. and Hinton, B.T., 2014. PTEN signaling through RAF1 proto-oncogene serine/threonine kinase (RAF1)/ERK in the epididymis is essential for male fertility. *Proceedings of the National Academy of Sciences*, 111(52), pp.18643-18648.
- Yang, Y., Sánchez-Tójar, A., O'Dea, R.E., Noble, D.W., Koricheva, J., Jennions, M.D., Parker, T.H., Lagisz, M. and Nakagawa, S., 2023. Publication bias impacts on effect size, statistical power, and magnitude (Type M) and sign (Type S) errors in ecology and evolutionary biology. *BMC biology*, 21(1), pp.1-20.

REVIEWERS' COMMENTS

Reviewer #1 (Remarks to the Author):

I thank the authors for carefully considering my points, and I'm happy with their adjustments to the manuscript. I'm still slightly skeptical about GSI as an index of sperm competition (because in many taxa the allometric slope is not one), but I doubt it matters very much for the authors' purposes.

Reviewer #2 (Remarks to the Author):

Dear authors,

I highly appreciate your efforts in addressing my previous comments. The inclusion of new analyses, coupled with the modifications and clarifications integrated into the text, has unquestionably enhanced the quality of the study. Thus, I have no further comments to add. Congratulations on this interesting study.

Reviewer #3 (Remarks to the Author):

For this revision, I think the authors have done a good job of better addressing the issue related to the impact of the percentage of lifespan covered by the various studies on the results of this meta-analysis. Throughout this new version, the analyses and discussions of this issue are much more prominent, which was the main problem in the previous version for me and is now corrected.

Apart from that, the authors have addressed all my various comments in this new version of the manuscript, so I don't have much to add here. I am particularly thankful that the authors took the time to perform an in-depth analysis of the impact of the researchers' goals for the various studies included on the results of this meta-analysis and to add this sensitivity analysis to the manuscript. Even if the effect found for this particular analysis is weak, I still think it will be informative for the readers.